

# Soil hydraulic material properties and subsurface architecture from time-lapse GPR

Stefan Jaumann[1,2] and Kurt Roth[1,3]

[1]Institute of Environmental Physics, Heidelberg University, Im Neuenheimer Feld 229, 69120 Heidelberg, Germany
[2]HGSMathComp, Heidelberg University, Im Neuenheimer Feld 205, 69120 Heidelberg, Germany
[3]Interdisciplinary Center for Scientific Computing, Heidelberg University, Im Neuenheimer Feld 205, 69120 Heidelberg, Germany

*Correspondence to:* Stefan Jaumann (stefan.jaumann@iup.uni-heidelberg.de)

**Abstract.** Quantitative knowledge of effective soil hydraulic material properties is essential to predict soil water movement. ground-penetrating radar (GPR) is a non-invasive and non-destructive geophysical measurement method to monitor the hydraulic processes precisely. Previous studies showed that the GPR signal from a fluctuating groundwater table is sensitive to the soil water characteristic and the hydraulic conductivity function. In this work, we show that this signal is suitable to

accurately estimate the subsurface architecture and the associated effective soil hydraulic material properties with inversion methods. Therefore, we parameterize the subsurface architecture, solve the Richards equation, convert the resulting water content to relative permittivity with the complex reflective index model (CRIM), and solve Maxwell's equations numerically. In order to analyze the GPR signal, we implemented a new heuristic event detection and association algorithm. Using events instead of the full wave regularizes the inversion as it allows to focus on the relevant measurement signal. Starting from an

ensemble of Latin hypercube drawn initial parameter sets, we sequentially couple the simulated annealing algorithm with the Levenberg–Marquardt algorithm. We apply the method to synthetic as well as measured data from the ASSESS test site and show that the method yields accurate estimates for the soil hydraulic material properties as well as for the subsurface architecture by comparing the results to references derived from time domain reflectometry (TDR) and subsurface architecture ground truth data.

# 1   Introduction

Quantitative understanding of soil water movement is in particular based on accurate knowledge of the subsurface architecture and the hydraulic material properties. As direct measurements are time-consuming and near to impossible at larger scales, soil hydraulic material properties are typically determined with indirect identification methods, such as inversion (Hopmans et al., 2002; Vrugt et al., 2008). Time domain reflectometry (TDR, e.g., Robinson et al., 2003) is a standard method to acquire the

required measurement data because it monitors the hydraulic processes accurately. Yet, being an invasive method, the TDR sensors disturb the soil texture of interest and typically require the maintenance of a local measurement station. Hence, it is difficult to apply the method at larger scales or to transfer the sensors to another field site. Ground-penetrating radar (GPR, e.g., Daniels, 2004; Neal, 2004) is an established non-invasive method for subsurface characterization and has the potential to





become a standard method for efficient, accurate and precise determination soil hydraulic material properties.

Available research studies regarding the estimation of hydraulic properties from GPR measurements may be categorized according to the applied methods for the different components of the research study, such as (i) GPR measurement procedure, (ii) experiment type, (iii) GPR simulation method, (iv) optimization method, and (v) evaluation method of the GPR signal.

Most of these studies either use on-ground, off-ground, or borehole GPR measurements. On-ground measurements (e.g., Buchner et al., 2012; Busch et al., 2012; Léger et al., 2015) are the easiest and the most flexible approach. They have the disadvantage, however, that the antenna characteristics is influenced by the coupling to the ground. Off-ground measurements (e.g., Lambot et al., 2009; Jadoon et al., 2012; Jonard et al., 2015) avoid these effects, but the measurements are influenced by surface roughness. Cross-borehole measurements allow for high resolution tomography of the subsurface (e.g., Ernst et al., 2007;

Looms et al., 2008; Scholer et al., 2011), but require boreholes which are destructive and expensive.

The applied experiment types range from infiltration, fluctuating groundwater table, to evaporation. Infiltration experiments (e.g., Léger et al., 2014; Thoma et al., 2014; Rossi et al., 2015) are fast (hours) and provide information about the near surface material properties. Through its dependence on the form of the infiltration front or plume, the resulting GPR signal can get rather complicated for quantitative evaluation. Difficulties arise from multiple reflections in the plume, waveguides in the

infiltration front, and from noise originating in small-scale heterogeneity or fingering). If the infiltration is done artificially, accurate knowledge of the spatial distribution of the infiltration flux is required. Also simultaneous GPR measurements during the infiltration process are difficult as the antenna coupling to the subsurface is influenced by the changing water content close to the surface. Fluctuating groundwater table experiments (e.g., Bradford et al., 2014; Léger et al., 2015) require intermediate time scales (hours to days) and provide information about the material properties close to the groundwater table. These exper-

iments are typically limited to fluvial or coastal areas or are induced artificially in test sites. Evaporation experiments (e.g., Moghadas et al., 2014) demand long time scales (weeks) as the hydraulic dynamics is slow at low water contents. Yet, this kind of experiment is important to understand the coupling of the pedosphere with the atmosphere quantitatively.

The applied models to simulate the GPR signal balance performance and accuracy. Ray tracing (Léger et al., 2014, 2015) is fast but merely yields an approximate solution of Maxwell's equations. These equations can be solved analytically with a

Green's function (e.g., Lambot et al., 2009; Busch et al., 2012; Jonard et al., 2015) assuming a layered subsurface architecture. Alternatively, Maxwell's equations can be solved numerically with the finite differences time domain (FDTD) method (e.g., Buchner et al., 2012). This method is computationally expensive, but grants full flexibility concerning the source wavelet and the subsurface architecture.

Due to the inherent oscillating nature of the electromagnetic signal, inversion of GPR data generally demands globally con-

vergent and robust optimization techniques. Sequentially coupling a globally convergent search algorithm (e.g., the global multilevel coordinate search algorithm (GMCS, Huyer and Neumaier, 1999) with the gradient-free locally convergent Nelder–Mead simplex algorithm (NMS, Nelder and Mead, 1965) was successfully applied to estimate hydraulic material properties from GPR measurements (e.g., Lambot et al., 2004; Busch et al., 2012; Moghadas et al., 2014). The NMS was further developed to the shuffled complex evolution (SCE-UA, Duan et al., 1992) which has become a standard tool in hydrology and was

also applied on GPR measurements (e.g., Léger et al., 2014, 2015; Jadoon et al., 2012). Additionally, Markov chain Monte





Carlo (MCMC) methods (e.g., Scholer et al., 2011; Thoma et al., 2014; Jonard et al., 2015) and data assimilation approaches (e.g., Tran et al., 2014; Manoli et al., 2015; Rossi et al., 2015) have been successfully applied so far.

The GPR signal has to be processed automatically for parameter estimation. Many full waveform inversion approaches directly use the resulting Green's function (e.g., Lambot et al., 2009; Busch et al., 2012; Jadoon et al., 2012) in the cost function. Using

the full antenna signal may lead to many local minima prohibiting a reliable identification of the global minimum (e.g., Bradford et al., 2014). In contrast, filtering the radargram with convolution approaches to determine travel time and amplitude of a limited number of events may even allow the application of efficient locally-convergent algorithms (e.g., Buchner et al., 2012). In homogeneous material, the transition zone above the groundwater table exhibits a smooth variation of the relative permittivity. As the resulting GPR reflection is a superposition of a series of infinitesimal contributions along the transition zone, the

detailed form of this reflection is sensitive to the variation of the relative permittivity. For simplicity, we refer to this reflection as transition zone reflection. Dagenbach et al. (2013) showed that this reflection is sensitive to the hydraulic material parameterization model. Bradford et al. (2014) measured the transition zone reflection of a drainage pumping test in a fluvial area with a antenna center frequency of 200 MHz and estimated hydraulic material properties. Klenk et al. (2015) employed numerical forward simulations and experiments using GPR antennas with higher antenna center frequency (400 and 600 MHz) for a more

detailed explanation of the transition zone reflection during imbibition, relaxation, and drainage. They also concluded that the transition zone reflection is sensitive on hydraulic material properties.

In this work, we use the transition zone reflection together with reflections at material interfaces to determine the subsurface architecture and the corresponding hydraulic material properties. Therefore, the ASSESS test site was forced with a fluctuating groundwater table ensuring large hydraulic dynamics. The time-lapse measurement data was acquired with a single channel

on-ground bistatic antenna operating at a center frequency of 400 MHz. Similar to Buchner et al. (2012), we solve Maxwell's equations in 2D and employ a new semi-automatic heuristic approach to extract travel time and amplitude of relevant reflections. This allows the optimization procedure to focus on the relevant information in the radargram and decreases the number of local minima. We draw an ensemble of initial parameter sets with the Latin hypercube algorithm. These parameter sets serve as initial parameters for the simulated annealing algorithm which is sequentially coupled with the Levenberg–Marquardt

algorithm. We show that this procedure allows to accurately estimate the subsurface architecture and the associated effective hydraulic material properties for synthetic and measurement data.

## 2   Methods

Parts of this section were already presented in Jaumann and Roth (2017) but are repeated here for the convenience of the reader.

### 2.1   ASSESS

The ASSESS test site is located near Heidelberg, Germany, and consists of three different kinds of sand (materials A, B, and C). Its effective 2D subsurface architecture is visualized in Fig. 1. The approximately $2\,\text{m} \times 20\,\text{m} \times 4\,\text{m}$ large site is equipped with a well to monitor and manipulate the groundwater table, a weatherstation, a tensiometer (UMS T4-191), as well as 32 soil



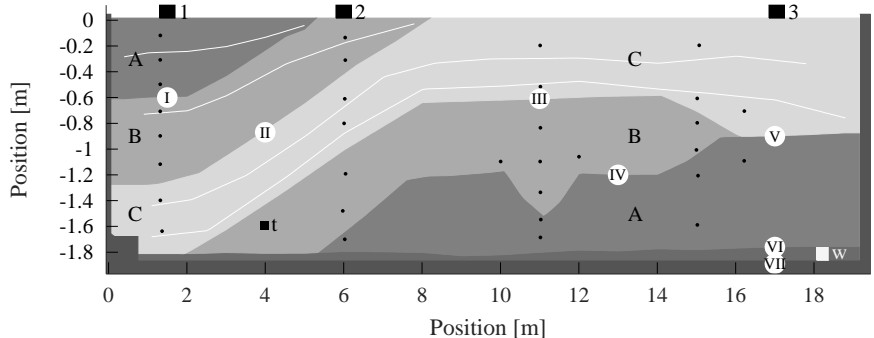

**Figure 1.** ASSESS emulates an effective 2D geometry with three distinct kinds of sand (A, B, and C). The hydraulic state can be manipulated with a groundwater well (white square at 18.3 m) and is monitored with three GPR antennas (1, 2, and 3), a tensiometer (black square, at 4.0 m), and 32 TDR sensors (black dots). The gravel layer at the bottom ensures a rapid water pressure distribution over the site. An L element (left wall, at 0.4 m) and compaction interfaces (white lines) were introduced during the construction. Additionally to those visualized, GPR evidence indicates additional compaction interfaces (Fig. 13). Roman numbers (I)–(VII) indicate material interfaces referred to in the text. Note the different scales on the horizontal and vertical axes.

temperature and TDR sensors. A geotextile separates the sand from an approximately 0.1 m thick gravel layer below, which ensures a rapid water pressure distribution and is the only connection of the groundwater well to the rest of the test site. Below this gravel layer, a basement layer partially consisting of reinforced concrete confines the site. As the test site is built into a former fodder silo, a concrete L element serves as additional wall. In order to stabilize the material during the construction, it was compacted. In addition to those shown in Fig. 1, GPR measurements indicate even more compaction interfaces (Fig. 13).

## 2.2 Representation

We follow Bauser et al. (2016) and define the *representation of a system* as a set consisting of: dynamics (mathematical description), subscale physics (material properties), forcing (superscale physics), and states.

### 2.2.1 Dynamics

The Richards equation (Richards, 1931),

$$\partial_t \theta - \nabla \cdot [K_\mathrm{w}(\theta)[\nabla h_\mathrm{m}(\theta) - \boldsymbol{e}_z]] = 0, \tag{1}$$

is the standard model to describe the propagation of the volumetric water content $\theta$ $(-)$ and the matric head $h_\mathrm{m}$ (m) in space and time $t$ (s). The solution of this partial differential equation requires the specification of material properties, namely the soil water characteristic $\theta(h_\mathrm{m})$ and the hydraulic conductivity function $K_\mathrm{w}(\theta)$, which are (i) highly non–linear, (ii) varying over many orders of magnitude, (iii) showing hysteretic behavior, (iv) impossible to determine a priori, and (v) very expensive to measure directly. The unit vector in $z$-direction $\boldsymbol{e}_z$ indicates the direction of gravity, typically pointing downwards.





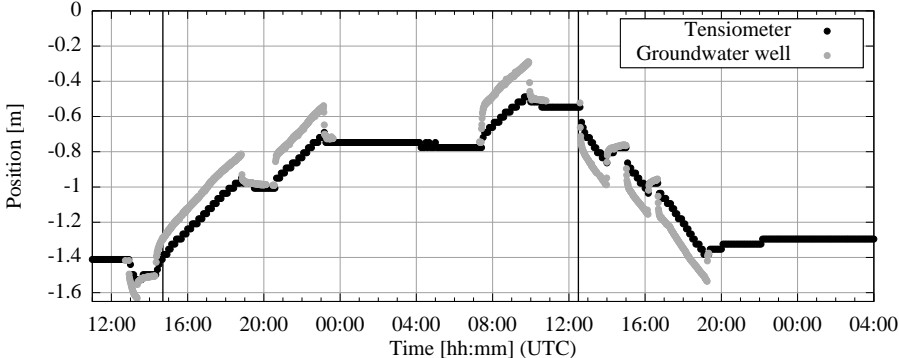

**Figure 2.** During the experiment with three distinct phases (initial drainage, multistep imbibition, and multistep drainage – separated by vertical lines), the position of the groundwater table was measured manually in the groundwater well and automatically with the tensiometer (Fig. 1). Notice that the difference between them is proportional to the driving force of water flow in the gravel layer.

### 2.2.2 Subscale physics

We choose the Brooks–Corey parameterization (Brooks and Corey, 1966) for the soil water characteristic $\theta(h_{\mathrm{m}})$, because it describes the materials in ASSESS appropriately (Dagenbach et al., 2013). Neglecting hysteresis, this parameterization may

be inverted for $\theta_{\mathrm{r}} \leq \theta \leq \theta_{\mathrm{s}}$, leading to

$$h_{\mathrm{m}}(\theta) = h_0 \left( \frac{\theta - \theta_{\mathrm{r}}}{\theta_{\mathrm{s}} - \theta_{\mathrm{r}}} \right)^{-1/\lambda} \tag{2}$$

including parameters representing a saturated water content $\theta_{\mathrm{s}}$ $(-)$, a residual water content $\theta_{\mathrm{r}}$ $(-)$, a scaling parameter $h_0$ (m) related to the air entry pressure $(h_0 < 0 \, \mathrm{m})$ and a shape parameter $\lambda$ $(-)$ related to the pore size distribution $(\lambda > 0)$.

Inserting the Brooks–Corey parameterization into the hydraulic conductivity model of Mualem (1976) yields the parameteri-

10 zation

$$K_{\mathrm{w}}(\theta) = K_0 \left( \frac{\theta - \theta_{\mathrm{r}}}{\theta_{\mathrm{s}} - \theta_{\mathrm{r}}} \right)^{\tau + 2 + 2/\lambda} \tag{3}$$

for the hydraulic conductivity function where $K_0$ $(\mathrm{m \, s^{-1}})$ is the saturated hydraulic conductivity and $\tau$ $(-)$ a fudge factor.

### 2.2.3 Forcing

The ASSESS site was forced with a fluctuating groundwater table leading to three characteristic phases (Fig. 2): (i) initial

drainage phase, (ii) multistep imbibition phase, and (iii) multistep drainage phase. We neglect evaporation in the following, because the experiment took place at the end of November and the weather was cloudy with 2–7 °C air temperature. The last precipitation was measured approximately 10 days before the experiment. More details about the experiment are given in Jaumann and Roth (2017). In this work, we only focus on the initial drainage and multistep imbibition phase.





### 2.2.4 State

The hydraulic state was monitored with GPR as well as with measurements of the position of the groundwater table in the groundwater well and at the position of the tensiometer. We used three shielded bistatic single channel $400\,\mathrm{MHz}$ GPR antenna pairs (Ingegneria dei Sistemi S.p.A., Italy). These antennas are referred to as antenna 1, 2, and 3, respectively. The measurement resolution was set to 2048 samples for 60 ns. In order to analyze the initial state of the test site, a multi-channel common offset measurement was acquired with antennas 1 and 2. The internal separation of the transmitter and receiver of these antennas is $0.14\,\mathrm{m}$. During the experiment, the antennas were used to measure three time-lapse radargrams. In this work, we focus on the quantitative evaluation of the time-lapse data from GPR antenna 3 (Fig. 1). These data are analyzed in detail in Sect. 3.3. Additionally, a mean soil temperature ($T_\mathrm{s} = 8.5\,^{\circ}\mathrm{C}$) and a mean direct current conductivity ($\sigma_\mathrm{dc} = 0.003\,\mathrm{S\,m^{-1}}$) was estimated from TDR related measurements available in ASSESS.

The observation operator required to compare the hydraulic state with the GPR measurement data involves the solution of the time-dependent Maxwell's equations in linear macroscopic isotropic media. These equations quantify the propagation of the electromagnetic field consisting of the electric field $\boldsymbol{E}$ and the magnetic field $\boldsymbol{B}$ (Jackson, 1999):

$$\nabla \times \frac{\boldsymbol{B}}{\mu} - \varepsilon \frac{\partial \boldsymbol{E}}{\partial t} = \sigma_\mathrm{dc} \boldsymbol{E} + \boldsymbol{J}, \tag{4}$$

$$\nabla \times \boldsymbol{E} + \frac{\partial \boldsymbol{B}}{\partial t} = 0. \tag{5}$$

The dielectric permittivity $\varepsilon = \varepsilon_0 \varepsilon_r$, magnetic permeability $\mu = \mu_0 \mu_r$, and direct current conductivity $\sigma_\mathrm{dc}$ are generally spatially variable and represent the electromagnetic properties of the subsurface. Here, we neglect dispersive effects ($\partial \varepsilon_r / \partial \omega = 0$) as well as the imaginary part of the dielectric permittivity ($\varepsilon_r \in \mathbb{R}$). The relative magnetic permeability is assumed to be that of vacuum ($\mu_r = 1$). The source current density $\boldsymbol{J}$ is applied at the position of the transmitter antenna.

The relative permittivity of the subsurface $\varepsilon_\mathrm{r} = \varepsilon_\mathrm{r,b}$ is calculated from the water content distribution $\theta$ resulting from the Richards equation using the complex refractive index model (CRIM) (Birchak et al., 1974):

$$\varepsilon_\mathrm{r,b}(\theta, T_\mathrm{s}, \phi)^\alpha = \theta \cdot \varepsilon_\mathrm{r,w}(T_\mathrm{s})^\alpha + (\phi - \theta) \cdot \varepsilon_\mathrm{r,a}^\alpha + (1 - \phi) \cdot \varepsilon_\mathrm{r,s}^\alpha, \tag{6}$$

with the geometry parameter $\alpha = 0.5$ (Roth et al., 1990). In order to apply the CRIM, the porosity $\phi$, the relative permittivity of water $\varepsilon_\mathrm{r,w}$, the relative permittivity of air $\varepsilon_\mathrm{r,a}$, and the relative permittivity of the soil matrix $\varepsilon_\mathrm{s}$ have to be known. The relative permittivity of air $\varepsilon_\mathrm{r,a}$ was set to 1. Assuming that the sand matrix consists mainly of Quartz ($SiO_2$) grains, the relative permittivity of the soil matrix $\varepsilon_\mathrm{r,s}$ was set to 5 (Carmichael, 1989). The porosity $\phi$ is assumed to be equal to the saturated water content $\theta_s$ (Eq. 2) which is estimated from the data. Following Kaatze (1989), we parameterize the dependency of the relative permittivity of water $\varepsilon_\mathrm{r,w}$ on the soil temperature $T_\mathrm{s}$ (°C) with

$$\varepsilon_\mathrm{r,w}(T_\mathrm{s}) = 10^{1.94404 - T_\mathrm{s} \cdot 1.991 \cdot 10^{-3}}. \tag{7}$$

In this work, the required direct current conductivity $\sigma_\mathrm{dc}$ of the subsurface is assumed to be constant in the whole architecture.




## 2.3 GPR Analysis

Similar to Buchner et al. (2012), we extract the signal travel time $t$ and the according amplitude $A$ at $M$ samples of the GPR
signal (events)

$$E_x \mapsto \left\{ \begin{pmatrix} t_1 \\ A_1 \end{pmatrix} \ldots \begin{pmatrix} t_M \\ A_M \end{pmatrix} \right\} \tag{8}$$

with a heuristic approach. This allows us to focus on the phenomena that are represented in the model and to exclude events
of, e.g., reflections originating from compaction interfaces or confining walls. However, this procedure demands an automatic
event association algorithm which associates events extracted from the measured signal with events extracted from the sim-
ulated signal. Thus, the evaluation method presented in this section consists of four steps: (i) signal processing, (ii) event
detection, (iii) event selection, and (iv) event association.

### 2.3.1 Signal processing

The GPR signal is processed for further evaluation according to the following steps: (i) time-zero correction, (ii) dewow filter,
(iii) 2D to 3D conversion, (iv) removal of the direct and trailing signal, and (v) normalization.
As the time-zero of the GPR antennas changes over time, we pick the direct signal and subtract it from the radargram for
time-zero correction. Then, a dewow filter is applied to subtract inherent low frequency wow noise of the GPR signal. Since
the observation is in 3D and the simulation in 2D, we convert the simulated signal to 2.5D, meaning to 3D with translational
symmetry perpendicular to the survey line and parallel to the ground surface (Bleistein, 1986). Note that ASSESS is built
accordingly (Sect. 2.1). In this work, the conversion is done for the frequency and the amplitude separately. First, each trace
is transformed to the frequency domain with the fast Fourier transform (FFT, denoted by $\widehat{\phantom{x}}$). Afterwards, the amplitude is
modified depending on the angular frequency $\omega$:

$$\widehat{A} \mapsto \widehat{A} \cdot \sqrt{\frac{|\omega_{k'}|}{2\pi}} \exp\left( -\frac{\mathrm{i}\pi}{4} \mathrm{sign}(\omega_{k'}) \right), \tag{9}$$

where i is the complex unit, $\omega_k = \Delta\omega \cdot (k' - \frac{K'}{2})$ ($k' \in \{1, \ldots, K'\}$, $K'$ is number of samples per trace enlarged to the next power
of two). Subsequently, all traces are transformed back to the time domain with the inverse FFT. Due to the frequency conversion
and the manipulation, a high frequency noise remains on the signal which is smoothed with a fourth order Savitzky–Golay
filter (e.g., Press, 2007, we employed the implementation of the 'signal' package for GNU Octave: https://octave.sourceforge.
io/signal/) using a window width of $41$ samples.
Accounting for energy dissipation in 3D requires additional manipulation of the amplitude. Assuming a direct ray path and
horizontal reflector with the reflector distance $d$ and mean square root of dielectric permittivity $\overline{\sqrt{\varepsilon}}$ along the ray path, this is
done via

$$A \mapsto A \sqrt{\frac{\overline{\sqrt{\varepsilon}}}{c_0 d}}. \tag{10}$$





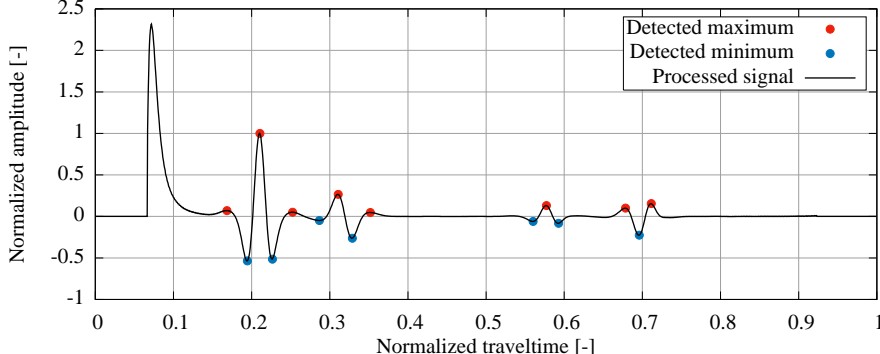

**Figure 3.** The amplitude of a trace is searched for extrema with a neighborhood search algorithm. For the subsequent evaluation, the amplitude of the detected events is normalized to the maximal absolute amplitude of all events detected in the trace. The direct signal and the trailing signal of the dewow filter are set to zero in a preprocessing step (Sect. 2.3.1) and events close to these signals are ignored.

Subsequently, the direct signal and the trailing signal of the dewow filter are set to zero. Finally, the each trace is normalized to its maximal absolute amplitude. Notice that the signal is renormalized later in the analysis of the GPR data (Sect. 2.3.4)

### 2.3.2 Event detection

To facilitate the identification of relevant events at large signal travel times, the normalized amplitude (original amplitude) is amplified quadratically with travel time. Subsequently, the extrema are detected with a local neighborhood search. Then, we keep a predefined number of events (15) with the largest amplified absolute amplitude. If the original amplitude of an detected extremum is below a predefined amplitude threshold (0.006) it is discarded in any case. In order to correct the perturbation in travel time due to the amplification and to cope with the discrete measurement resolution, we fit a Gaussian centered at the

travel time of the detected event with width of $\pm 5$ samples to the original amplitude data. The resulting amplitude and travel time of the extremum are used for the following evaluation.

### 2.3.3 Event selection

After the event detection, the measured signal is inspected manually together with the detected events. In this one-time preprocessing step events can either be deleted or added manually. This ensures that only those events enter the parameter estimation

that are also represented in the model. This step is skipped for the analysis of the simulated data.

### 2.3.4 Pairwise event association

The selected events extracted from the measured data have to be associated with the detected extracted from the simulated data for the parameter estimation. Therefore, Buchner et al. (2012) tested all possible combinations of events, using the one with the minimal summed absolute travel time difference. However, this is only feasible for a small number of events. As we



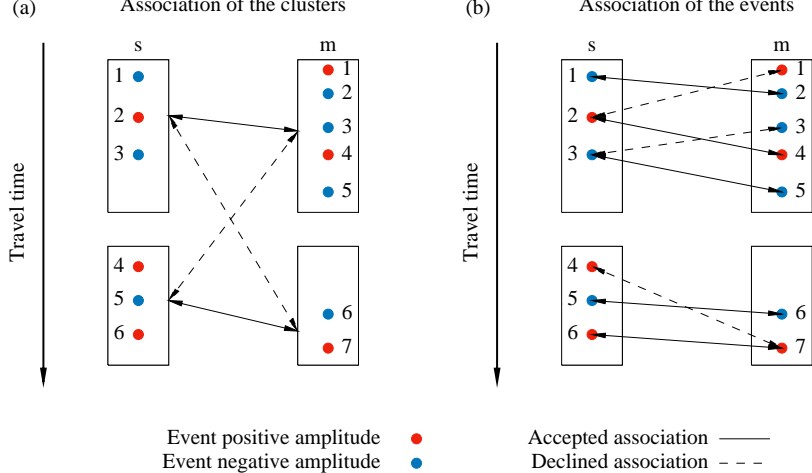

Event positive amplitude ● Accepted association ———
Event negative amplitude ● Declined association – – –

**Figure 4.** Exemplary association of simulated (s) and measured (m) events with indices 1–6 and 1–7, respectively. The color of the dots indicates the sign of amplitude of the events. (a) The detected events (Fig. 3) are aggregated in clusters to minimize the number of possible event combinations. The clusters are associated such that the summed absolute travel time difference of the mean travel time of the events in the cluster is minimal. (b) The events in the clusters are associated according to consistent temporal order and amplitude sign. Hence, if $(t_{s,1}, A_{s,1})$ is associated with event $(t_{m,2}, A_{m,2})$, event $(t_{s,2}, A_{s,2})$ can not be associated with event $(t_{m,1}, A_{m,1})$, if $t_{m,1} < t_{m,2}$ or $\text{sign}(A_{s,2}) \neq \text{sign}(A_{m,1})$. Solid (dashed) arrows indicate some of the accepted (declined) association combinations. The combination with maximal number of associations and minimal summed absolute travel time difference is used for evaluation.

are not using a Gaussian convolution of the data but the data themselves, the number of events increases. Hence, testing all combinations is often prohibitively expensive. In order to exclude combinations a priori, the detected events are aggregated in clusters (Fig. 4a). Then, these clusters are associated by testing all possible combinations. We use the combination with the minimal summed absolute travel time difference. Afterwards, the events aggregated in the associated clusters are associated

5  themselves. The applied association procedure requires the events to have an identical amplitude sign and a consistent temporal order which reflects the principle of causality (Fig. 4b). We iterate over all according combinations to find the association with the maximal number of associated events and the minimal summed absolute travel time difference. It is critical to also consider combinations where some intermediate events (e.g., $(t_{s,2}, A_{s,2})$ in Fig. 4) can not be associated. After the association of the events, outliers are detected by calculating the mean and standard deviation of the travel time differences. All associations

10  are discarded which exhibit an absolute travel time difference larger than 3 standard deviations of all absolute travel time differences. Finally, the amplitude of the associated events is normalized to the maximal absolute amplitude of the associated events in each trace.





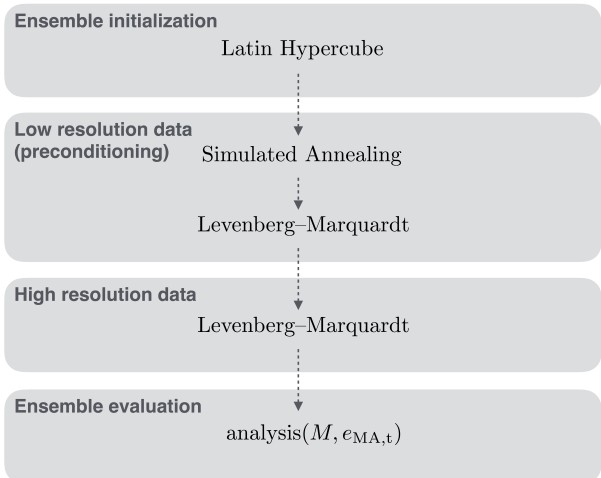

**Figure 5.** We choose an iterative parameter estimation procedure which (i) allows to minimize the computational cost and (ii) facilitates the implementation of tagging (Sect. 2.4.1). Therefore, we precondition the Latin hypercube sampled parameter sets with a data set with reduced number of traces (low resolution data) and use optimization algorithms which compare the parameter sets sequentially. The preconditioned parameter sets for each ensemble member serve as initial parameters for the final parameter estimation based on high resolution data. The subsequent evaluation of the ensemble is based on the number of associated events $M$ and the mean absolute error in travel time $e_{\mathrm{MA,t}}$ (Sect. 3.1.2).

## 2.4 Parameter estimation

Inversion of GPR data typically requires globally convergent parameter estimation algorithms which are computationally expensive. In order to keep the parameter estimation procedure efficient, we use an iterative strategy (Fig. 5). We start the optimization procedure by drawing an ensemble of initial parameter sets with the Latin hypercube algorithm (implemented by the pyDOE package, https://github.com/tisimst/pyDOE). The most expensive operation of the forward simulation is the calcu-

5   lation of the observation operator, which includes the solution of Maxwell's equations (Sect. 2.2.4) and the subsequent event association (Sect. 2.3). Hence, as time-lapse GPR data are highly correlated in experiment time (e.g., Fig. 13), we equidistantly subsample the number of traces of the time-lapse GPR radargram and generate a data set with lower temporal resolution. We use those data to improve the the distribution of the initial parameters (preconditioning). Therefore, the drawn parameter sets are used to initialize the simulated annealing algorithm (Sect. 2.4.2) which allows for a robust, fast, and easy to implemented

10  parameter update. Subsequently, the resulting parameters serve as initial parameters for the Levenberg–Marquardt algorithm (Sect. 2.4.3) concluding the preconditioning step. The preconditioned parameter sets are used as the initial parameter sets for the more expensive optimization of high resolution data set with the Levenberg–Marquardt algorithm. The details of the setup and the analysis of the parameter estimation are given in Sect. 3.1.2.



### 2.4.1 Objective function

Assuming $P$ parameters $p_\pi$ and $M$ associations of measured events $(t_{\mu,\mathrm{m}}, A_{\mu,\mathrm{m}})$ with simulated events $(t_{\mu,\mathrm{s}}(\boldsymbol{p}), A_{\mu,\mathrm{s}}(\boldsymbol{p}))$, the $\chi^2$ objective function is given by

$$\chi^2(\boldsymbol{p}) = \frac{1}{2}\sum_{\mu=1}^{M}\left(\frac{t_{\mu,\mathrm{s}}(\boldsymbol{p}) - t_{\mu,\mathrm{m}}}{\sigma_{t,\mu}}\right)^2 + \left(\frac{A_{\mu,\mathrm{s}}(\boldsymbol{p}) - A_{\mu,\mathrm{m}}}{\sigma_{A,\mu}}\right)^2 = \frac{1}{2}\sum_{\mu=1}^{M} r_{t,\mu}^2 + r_{A,\mu}^2 \tag{11}$$

with the constant standard deviation of the measured normalized travel times $\sigma_{t,\mu} = \sigma_t$ and of the measured normalized am-
plitudes $\sigma_{A,\mu} = \sigma_A$ leading to the residuals in travel time $r_{t,\mu}$ and amplitude $r_{A,\mu}$. Due to the oscillating nature of the GPR signal and due to the analysis (Sect. 2.3), the $\chi^2$ function is not convex and may even be discontinuous at some points, as the number of associated events $M$ is not necessarily constant during the minimization process. To compensate for adding and removing simulated or measured events, Buchner et al. (2012) introduced *tagging*. If the number of measured events is smaller than the number of the simulated events, the simulated events that are not associated are excluded. Alternatively, if there are more measured events, measured events without partner are tagged as partnerless. If a reflection event has been tagged and becomes untagged after the parameter update, the contribution of the event and its new partner to the objective function is added to the previous objective function value. If an event has not been tagged and becomes tagged after the parameter update, the contribution to the cost function is subtracted from the previous objective function value.

### 2.4.2 Simulated annealing

We choose the simulated annealing algorithm (Press, 2007) to start the minimization of the objective function (Eq. 11), because this algorithm is gradient-free and updates the parameters statistically. Additionally, tagging (Sect. 2.4.1) can be implemented easily and it also allows uphill steps, which can be favorable if the events are not yet associated to their appropriate reflection. If the parameter update is drawn from the whole parameter space, the algorithm is globally convergent. However, this approach is typically inefficient. We mainly use the simulated annealing algorithm to find a parameter set that associates the events to their appropriate reflection such that the more efficient gradient-based algorithm can take over. Hence, we search the neighborhood for better parameters starting from Latin hypercube sampled initial parameters $\boldsymbol{p}_{\pi,0}$. For each iteration $i$ $(1,\dots,I)$, new parameters are proposed randomly via

$$\boldsymbol{p}_{\pi,i+1} = \boldsymbol{p}_{\pi,i} + m \cdot (\boldsymbol{p}_{\pi,\max} - \boldsymbol{p}_{\pi,\min}) \cdot u_{\mathrm{p}}, \tag{12}$$

with a mobility parameter $m = 0.1$, uniformly distributed random number $u_{\mathrm{p}} \sim \mathcal{U}(-1,1)$, and the parameter limits $\boldsymbol{p}_{\pi,\max}$ and $\boldsymbol{p}_{\pi,\min}$. In order provide the control parameter $T$, which is an analogue of temperature, we choose an exponential cooling schedule

$$T_{i+1} = T_0 \cdot \alpha^{i+1}, \tag{13}$$



with $\alpha = 0.85$ and initial temperature $T_0 = 10^3$ which is of the order of the initial cost function value. According to Metropolis et al. (1953), we draw an uniformly distributed random number $u_{\mathrm{d}} \sim \mathcal{U}(0,1)$, calculate the acceptance probability

$$P_{i+1} = \exp\left(-\frac{\chi_{i+1}^2 - \chi_i^2}{k \cdot T_{i+1}}\right), \tag{14}$$

choosing parameter $k = 1$, and accept the proposed parameter set if $P_{i+1} > u_{\mathrm{d}}$ or else draw a new parameter set.

### 2.4.3 Levenberg–Marquardt

The Levenberg–Marquardt algorithm is implemented as described by Jaumann and Roth (2017). However, in order to suc-
cessfully apply this gradient-based algorithm to GPR data, the optimization has to be regularized. Therefore, we focus with this algorithm in particular on the improvement of small residuals, because if the small residuals improve, the larger residuals are likely to also improve in subsequent iterations due to the temporal correlation of the data. Therefore, we tag events with $r_{t,\mu} > 100$ or $r_{A,\mu} > 100$. Tagged events are excluded from the optimization by setting the according entries in the Jacobi matrix ($J_{\mu,\pi} = \partial r_\mu / \partial p_\pi$) to zero. The event association may also change during the perturbation of the parameters for the
numerical assembly of the Jacobi matrix. This can lead to large changes in the residuals, which in turn may lead to a disturbed parameter update. Hence, corresponding entries of large changes in the residual $\mathrm{abs}\left(r_\mu(\boldsymbol{p}_{\mathrm{perturbed}}) - r_\mu(\boldsymbol{p})\right) > 50$ are also set to zero together with entries of the Jacobi matrix that are larger than $10^4$.

We choose $\lambda_{\mathrm{initial}} = 5$ as initial value for $\lambda$ and apply the delayed gratification method by decreasing (increasing) $\lambda$ by a factor of 2 (3) if the parameter update is successful (not successful). This assures that the algorithm takes small steps such that
association and the Jacobi matrix can adapt smoothly.

## 3  Application

In this section, we apply the presented methods to the acquired GPR data. Therefore, we first explain the setup of the case study and the parameter estimation procedure (Sect. 3.1.1). Then, we test the method with synthetic data to understand the phenomenology of the data and capabilities of the method (Sect. 3.2). Finally, we apply the method to the measured data (Sect.
3.3) and analyze the accuracy of the resulting parameters.

### 3.1  Setup of the case study

### 3.1.1  Implementation

The numerical solution of the Richards equation (Eq. 1) is based on $\mu\varphi$ (muPhi, Ippisch et al., 2006) which uses a cell centered finite volume scheme with full upwinding in space and an implicit Euler scheme in time. The nonlinear equations are
linearized by an inexact Newton method with line search and the linear equations are solved with an algebraic multigrid solver. We solve Richards equation in 1D ($z$ dimension) on a structured grid with a resolution of $\approx 0.005$ m. Generally, the boundary condition is implemented with a Neumann no-flow condition. However, during the forcing phases, we prescribe the measured




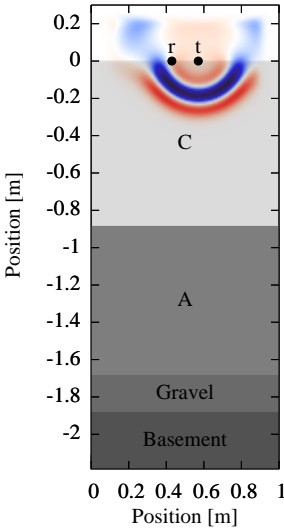

**Figure 6.** For the simulation of the GPR signal of antenna 3, we assume a layered subsurface architecture (Fig. 1). The transmitter of the antenna is represented with an infinitesimal dipole (t) and the electric field is read at the position of the receiver antenna (r). An absorbing layer is used as boundary condition.

groundwater table as a Dirichlet boundary condition at the position of the groundwater well. We initialize the simulation with hydraulic equilibrium based on the measured groundwater table position. The simulated water content is converted to relative

permittivity via the CRIM using the mean soil temperature $T_s = 8.5\,°C$ (Sect. 2.2.4).

To simulate the temporal propagation of the electromagnetic signal, we solve Maxwell's equations (Sect. 2.2.4) in 2D with the MIT electromagnetic equation propagation software (MEEP, Oskooi et al., 2010). The transmitter antenna is represented with an infinite dipole pointing in $x$ dimension. Thus, we neglect any effects from the real antenna geometry (bow tie), cross coupling or antenna shielding. The antenna source current density $\boldsymbol{J}$ is given by a Ricker excitation function (first derivative of

5 a Gaussian–shaped function) with a center frequency of $400\,\mathrm{MHz}$. The receiver antenna is not represented explicitly. Instead, $E_x$ is read directly at the position of the receiver antenna. We use the antenna separation of the real GPR system ($0.14\,\mathrm{m}$) in the simulation. Perfectly matched layers (PML) of $0.15\,\mathrm{m}$ thickness serve as boundary condition. The initial electromagnetic field in the domain is zero. We use one tenth of the minimal wavelength $\lambda_{w,\min}$ as upper limit for the spatial resolution $\Delta z$:

$$\Delta z \leq \frac{\lambda_{w,\min}}{10} = \frac{\frac{c_0}{\sqrt{\varepsilon_{r,\max}}}}{10 f_{\max}} \approx 0.007\,\mathrm{m}, \tag{15}$$

10 with the speed of light in vacuum $c_0$, maximal frequency $f_{\max} = 2 \cdot 400\,\mathrm{MHz}$, and $\varepsilon_{r,\max} = 31.25$ corresponding to $\theta_{s,\max} = 0.5$. Hence, we choose the numerical resolution $\Delta z = 0.005\,\mathrm{m}$ for the 2D isotropic, structured, and rectangular grid. Therefore, the simulated one dimensional relative permittivity distribution is extruded in the $y$ dimension. The Courant number for the FDTD




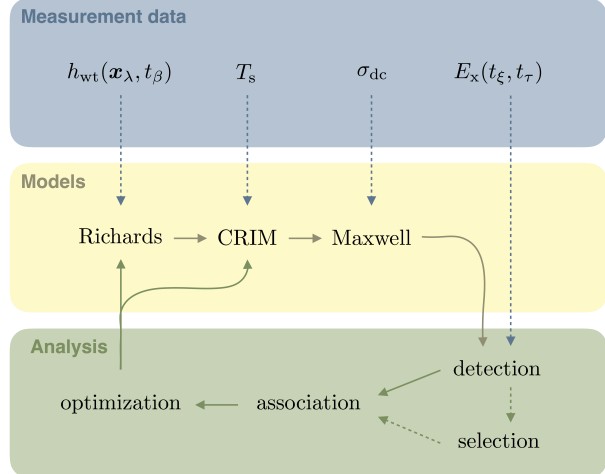

**Figure 7.** The available hydraulic potential $h_{\mathrm{wt}}$ is measured at the position of the groundwater well $\boldsymbol{x}_\lambda$ times $t_\beta$. These measurements are used as a boundary condition for the Richards equation (Sect. 2.2.1). Estimates for the soil temperature $T_{\mathrm{s}}$ and the direct current conductivity $\sigma_{\mathrm{dc}}$ are derived from TDR related measurements. The actual signal of the GPR system is proportional to $x$ component of the electric field $E_x$ and measured discretely at experiment time $t_\xi$ and signal travel time $t_\tau$. This signal is used for event detection and event selection (Sect. 2.3). The simulated water content distribution is converted to relative permittivity distribution with the CRIM and used to solve Maxwell's equations (Sect. 2.2.4 and 3.1.1). After event detection, the simulated events are assigned to measured events. The mapping of the events is used to calculate objective function value during the optimization step (Sect. 2.4). Dashed arrows indicate initial preprocessing steps, whereas solid arrows indicate iterative steps required for the optimization.

method is set to $0.5$.

To avoid multiple reflections at the air-soil boundary, we set the relative permittivity above the soil to $3.5$, which is typical for dry sand. This is justified, as no evaluation of air wave or ground wave is done and the amplitude is normalized according to the detected events. The permittivity of the basement below ASSESS is set to $23.0$, based on previous simulations. The direct current conductivity of the subsurface $\sigma_{\mathrm{dc}}$ is set to $0.003\ \mathrm{Sm}^{-1}$ (Sect. 2.2.4). All electromagnetic properties are smoothed by MEEP according to Farjadpour et al. (2006). The subsurface architecture is represented with layers. The position of these layers is parameterized and can be estimated. For illustration, the setup is shown in Fig. 6.

### 3.1.2 Setup of the parameter estimation

General setup of the optimization is explained with Fig. 7. This setup is used in an iterative approach (Fig. 5), where we selected ever fewer of the traces of the time-lapse GPR data to generate a data set with decreased temporal resolution. The data set with high (low) resolution includes $86$ $(9)$ traces corresponding to one trace per $15$ $(150)$ min. We draw $40$ initial parameter sets with the Latin hypercube algorithm within the sample range given in Table 1 and use the data set with low temporal resolution to improve these parameter sets. Therefore, we run $200$ iterations with the simulated annealing algorithm (Sect. 2.4.2). Notice that



the parameter fit range given in Table 1 determines the parameter update via $p_{\pi,\max}$ and $p_{\pi,\max}$ (Eq. 12). After the simulated annealing algorithm, we run maximally 15 iterations of the Levenberg–Marquardt algorithm (Sect. 2.4.3) which completes the precondition step. The resulting parameter sets serve as initial parameters for the Levenberg–Marquardt algorithm which is applied to the data with high temporal resolution.

In order to evaluate the performance of the ensemble members, we use the mean absolute error in travel time $e_{\mathrm{MA,t}}$, because this statistical measure is independent of the number of associated events. These are accounted for by choosing those 10 members with minimal $e_{\mathrm{MA,t}}$ where at least $85\%$ of the measured events are associated. Each of these members has locally optimal parameters. However, the exact position of these local minima typically depends on (i) the settings of the optimization

algorithms, (ii) the choice of the events to be evaluated, and (iii) the random numbers drawn in the simulated annealing algorithm. There is also no guarantee that the global optimum was found by one of the ensemble members. However, the distribution of these 10 best ensemble members contains valuable information about the shape of the $\chi^2$ surface. To account for this information, we (i) analyze the mean parameter set of the best members and (ii) use the according standard deviation to indicate the uncertainty of these parameters. Notice that the mean parameter set is not necessarily optimal. However, if

uncertainty on the parameters is small, this result is typically more reliable results than the best ensemble member.

The standard deviation of the measured data, $\sigma_t \approx 6 \cdot 10^{-4}$ and $\sigma_A \approx 5 \cdot 10^{-3}$ for normalized travel times and amplitudes is used as the standard deviation of the residuals in the objective function (Sect. 2.4.1).

## 3.2   Synthetic data

### 3.2.1   Phenomenology

The phenomenology of the transition zone reflection for characteristic times during imbibition, relaxation, and drainage was discussed by Klenk et al. (2015) for typical coarse sand. Here, we focus on the temporal development of this reflection during imbibition and equilibration. Therefore, we simulated water content distribution in the 1D profile located at $17.05\,\mathrm{m}$ of ASSESS using parameters typical for coarse-textured sandy soils (Table 2). The results are visualized over time (Fig. 8a) and over the water content (Fig. 8b). Initialized with hydraulic equilibrium, the simulation starts with the initial drainage step (Sect. 2.2.3)

where the groundwater table is lowered. Hence, the material with high initial water content is desaturated. After the subsequent equilibration step, the groundwater table is raised during the subsequent imbibition step. Generally, the Brooks–Corey parameterization (Eq. 2) features a sharp kink where air enters the material at the upper end of the capillary fringe. Additionally, the imbibition causes another kink in the water content distribution (at marker (2) in Fig. 8b), because the relaxation time from the hydraulic non-equilibrium is much shorter at high water contents compared to the relaxation time at low water contents. This

is due to the strong non-linear dependency of the hydraulic conductivity (Eq. 3) on the water content leading the differences in hydraulic conductivity of several orders of magnitude. Hence, the transition zone is sharpened during the imbibition. During the equilibration step after the first imbibition, the transition zone smoothes. Thus, the water content increases in the material with low water content (3) and decreases in the material with high water content (4). This smoothing process strongly depends on both the soil water characteristic and the hydraulic conductivity function. Sharpening and smoothing of the transition zone





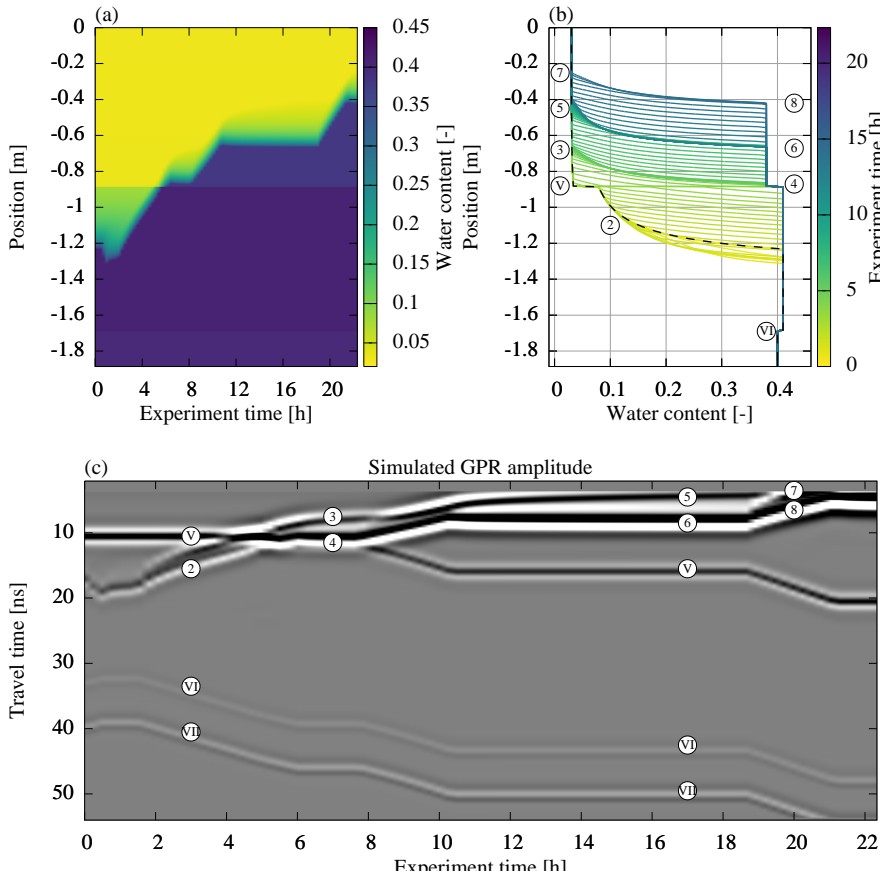

**Figure 8.** We used hydraulic parameters representing coarse-textured sandy soils (Table 2) to generate the synthetic data for parameter estimation according to Sect. 3.1.1. Subfigure (a) shows the simulated water content in color code over experiment time, whereas subfigure (b) shows the same data in a line plot emphasizing temporal development of the water content distribution. The initial water content distribution is marked with a black dashed line. Subfigure (c) shows the according simulation of the GPR signal. The imbibition leads to a characteristic transition zone reflection (2). The temporal evolution of this reflection is sensitive on the initial water content distribution, the soil water characteristic and the hydraulic conductivity function. Except for the normalization, the data are processed according to Sect. 2.3, including a dewow filter and 2D to 3D conversion. In contrast to the quantitative evaluation, the radargram is normalized to the maximal absolute amplitude, facilitating the visual comparison of the traces. The markers are used consistently in this paper and are further explained in the text.





are repeated consistently for the other subsequent imbibition and equilibration phases (5), (6) and (7), (8).

According to the CRIM (Sect. 2.2.4), the relative permittivity distribution has the same shape as the water content distribution. Hence, kinks in the water content distribution directly induce partial reflections of the GPR signal (Fig. 8c). Shortly after starting the imbibition, the amplitude of the reflection at the additional kink (2) increases. After passing the material interface (V), the spatial distance of the kinks increases such that the two resulting reflection wavelets (3) and (4) are separable. The signal in between these wavelets is a superposition of infinitesimal reflections which contain detailed information about the form of the transition zone. Notice that the reflection (3) scans the initial water content distribution, which in steady state corresponds to the soil water characteristic. With progressing equilibration, the amplitude of reflection (3) decreases as the transition zone

5    smoothes. The GPR signal of the subsequent imbibition and equilibration phases (5), (6) and (7), (8) show similar behavior and emphasize the relatively long time scale for hydraulic equilibration of sandy materials.

In summary, this numerical simulation confirms qualitatively (i) that the dynamics of the fluctuating groundwater table is sensitive to both the soil water characteristic and the hydraulic conductivity function and (ii) that the transition zone reflection leads to tractable reflections during the imbibition step.

## 10   3.2.2   Results and discussion

The resulting soil water characteristics for material A (Fig. 9a) exhibit a similar curvature but are shifted. Both the parameters $h_0$ and $\lambda$ influence the shape of the desaturated transition zone. Hence, merely evaluating the shape of the desaturated part of the transition zone is not necessarily sufficient to uniquely identify both parameters leading to large correlation coefficients. However, parameter $h_0$ additionally determines the extent of the capillary fringe. If the evaluation is also sensitive on the extent of the capillary fringe, $h_0$ can be uniquely identified which significantly decreases the correlation between $h_0$ and $\lambda$. Hence, we conclude that the strong correlation of the parameters $h_{0,\mathrm{C}}$ and $\lambda_\mathrm{C}$ ($-0.7$, Fig. 10) indicates that the evaluation is more sensitive to the shape of the desaturated part of the transition zone than to the extent of the capillary fringe.

As the architecture is a layered structure where material C is located above material A (Fig. 6), the water content in material C contributes to the travel time of the other reflections. This introduces correlations of all the parameters associated with the soil water characteristic of material C to $\theta_{s,\mathrm{A}}$. A high correlation of parameters indicates that the problem is not well-posed. This typically increases the number of local minima and thus the uncertainty of the parameters.

The saturated hydraulic conductivity of material A (Fig. 9b) is approximately one order of magnitude smaller than the saturated

hydraulic conductivity of material C. As the 1D architecture is forced at the lower boundary, the hydraulic conductivity of material A limits the water flux into material C. Hence, the data are not sensitive on $K_{s,\mathrm{C}}$. Yet, the uncertainty of the hydraulic conductivity decreases for low water contents as the reflection at the additional kink (Sect. 3.2.1) is sensitive to the hydraulic conductivity. The hydraulic conductivity function (Eq. 3) is not unique if $K_s$ is not fixed. This leads to a strong correlation of the parameters $K_{s,\mathrm{C}}$ and $\tau_\mathrm{C}$ (0.6, Fig. 10). Note that the uncertainty of the saturated hydraulic conductivity of material A also

influences the uncertainty of the hydraulic conductivity of material C.

The uncertainty in the soil water characteristic of material C (Fig. 9c) is largest for low water contents, as there are only few data points available. In particular, this increases the uncertainty of $\lambda_\mathrm{A}$ ($\pm0.7$, Table 2). The material properties of the unsaturated





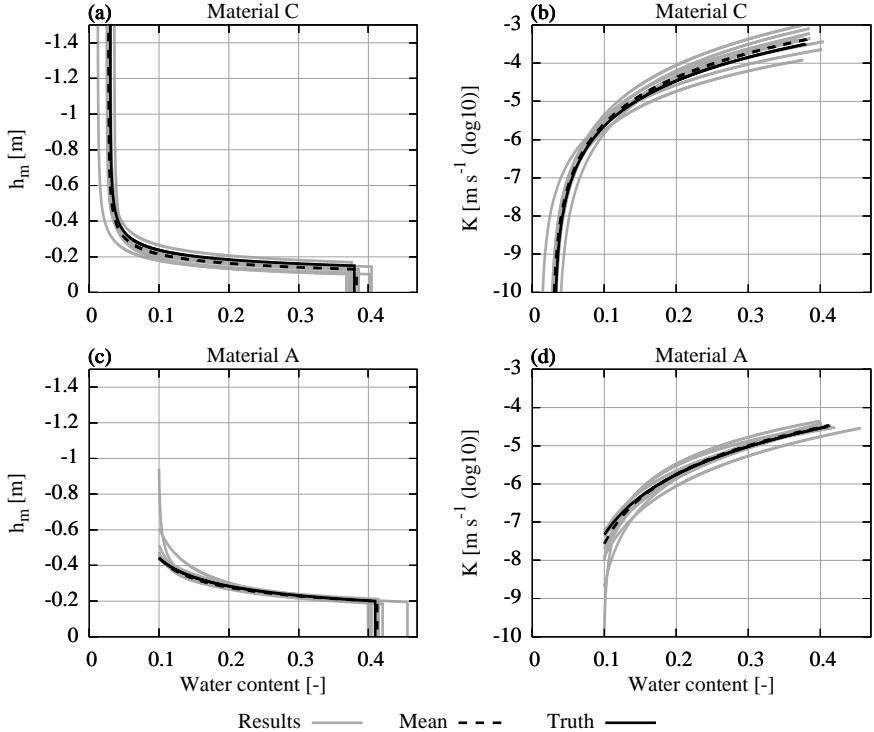

**Figure 9.** The resulting material parameters estimated from synthetic data are shown for the 10 best ensemble members (Sect. 3.1.2) together with the mean of these parameter sets and the true parameter set (Table 2). The plot range of the parameters is adjusted to the water content range of the data.

material A are only monitored during the first $\approx 5$ h of the experiment and are independent of the largest part of the other data. This regularizes the optimization leading to fewer local minima. Similar to material C, the parameters $h_{0,A}$ and $\lambda_A$ are strongly

correlated ($-0.6$). Yet, the uncertainty in $h_{0,A}$ is relatively small ($\pm 0.008$, Table 2) mainly because it is essentially uncorrelated to other parameters. In contrast, the parameter $\theta_{s,A}$ is correlated to the parameters $h_{0,C}$, $\lambda_C$, $\theta_{s,C}$, and $\theta_{r,C}$ as wrong parameters for material C introduce changes in the total water content which can be partially balanced out by adjusting $\theta_{s,A}$.

The uncertainty of the saturated hydraulic conductivity of material A (Fig. 9d) is comparably small as the largest fraction of the data are influenced by this parameter. Hence, the parameters $\tau_A$ and $K_{s,A}$ are only very weakly correlated.

The correlation coefficients (Fig. 10) also show that both the permittivity and the thickness of the gravel layer can be estimated reliably with the presented evaluation method using travel time and amplitude information of a single channel. Evaluation methods that merely exploit the signal travel time (e.g., Gerhards et al., 2008), require additional measurements to achieve this goal.

In order to further investigate the quality of the mean parameter set, we simulated the resulting water content distribution

(Fig. 11a) and calculated the difference to the true water content distribution (Fig. 11b). Due to the narrow pore size distribution





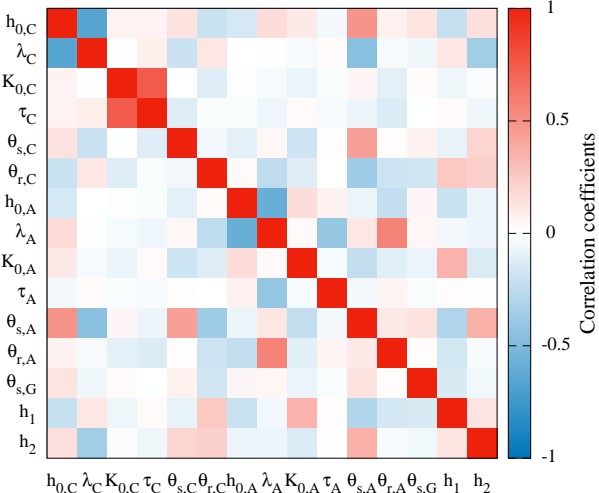

**Figure 10.** The correlation coefficients for the mean parameter set are analyzed in detail in the text. Notice that the porosity of the gravel ($\theta_{s,G}$) as well as the position of the material layers ($h_1$ and $h_2$) can be reliably estimated from single channel GPR when evaluating both the signal travel time and amplitude.

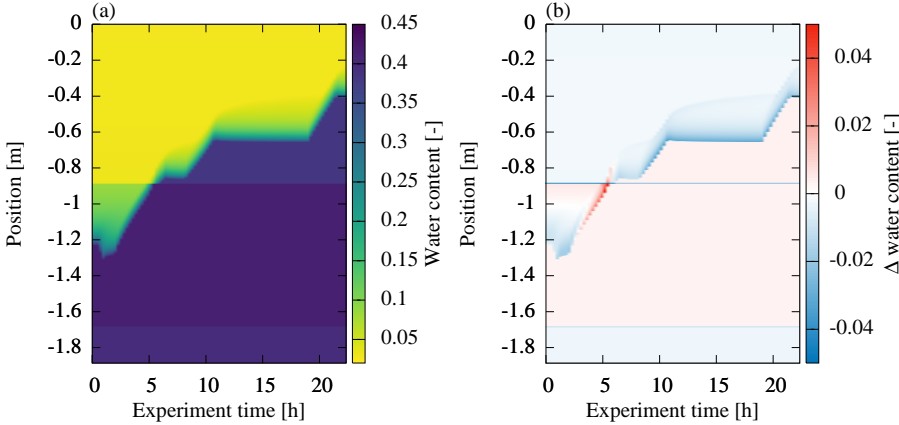

**Figure 11.** We calculated (a) the resulting water content distribution of the mean parameter set and (b) the difference to the true water content distribution (Fig. 8). The mean absolute deviation of the volumetric water content is $0.004$. The over all balance of the volumetric water content can be characterized by calculating the mean of the summed difference per grid cell over all measurement times which yields $-0.003$. Hence, the mean parameter set generally underestimates the water content in the profile.





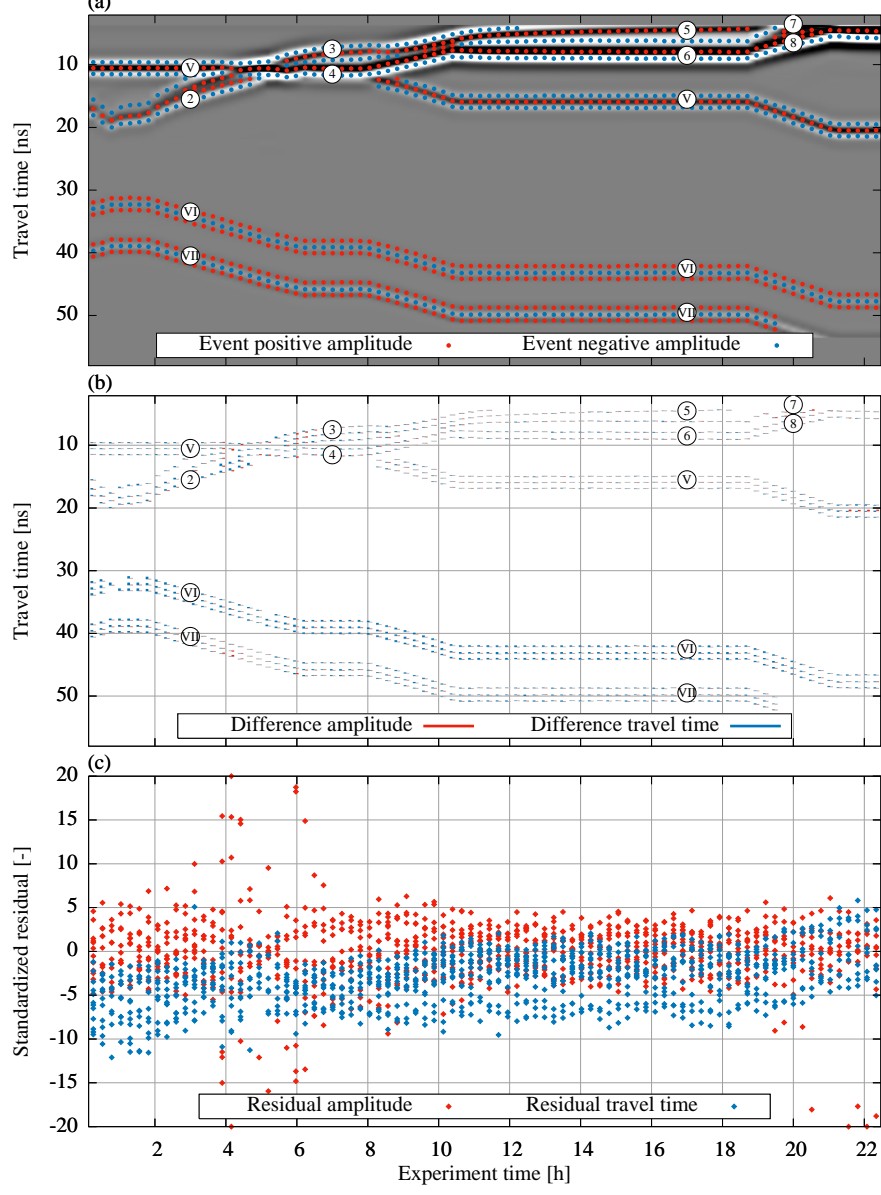

**Figure 12.** The evaluation the synthetic GPR data is separated in three parts: Subfigure (a) shows the detected (Sect. 2.3.2) and selected (Sect. 2.3.3) events which are used as synthetic measurement data. Except for the normalization, the data are processed according to Sect. 2.3, including a dewow filter and 2D to 3D conversion. The radargram is normalized to the maximal absolute amplitude, facilitating visual comparison of the traces. Subfigure (b) shows resulting differences in travel time and amplitude of the mean parameter set. The differences of the amplitude are given in arbitrary units which are consistent for the whole work. Subfigure (c) shows standardized residuals of the differences, essentially zooming into the small differences given in subfigure (b). Note that outliers are set onto the boundary. The markers for the reflections are used consistently in this paper.



of the sandy material, small deviations in the parameters $h_0$ and $\lambda$ lead to large differences in the volumetric water content above the capillary fringe ($\approx \pm 0.04$). Combined with deviations in the position of the material interface, the largest differences in volumetric water content reach up to $0.17$. Still, the mean absolute deviation of the volumetric water content is $0.004$.

These deviations also cause residuals in the GPR signal (Fig. 12), which are most evident for the reflection at the gravel layer (VI). The bias of its travel time shows that the total water content above the gravel layer is underestimated with the mean parameter set. This bias is essentially balanced out with the properties of the gravel layer. However, the reflection originating from the basement of ASSESS (VII) reveals residuals that decrease as soon as the groundwater table is above the initial groundwater table. This indicates (i) deviations in the initial water content distribution and (ii) that the hydraulics during the initial drainage phase is not correct.

Similar to the analysis of the deviation in water content (Fig. 11), the largest residuals in unsaturated hydraulics are found where the groundwater table is crossing the interface of materials A and C. This indicates that the interference of the according reflections still contains information which could not be exploited in the parameter estimation procedure. Apart from that, the deviations in the material properties of unsaturated material C do not lead to significant residuals in the GPR signal, although the deviations in water content are considerable.

### 3.3 Measured data

#### 3.3.1 Phenomenology

The common offset GPR measurement (Fig. 13a) reveals the initial state of ASSESS. The reflections of the material interfaces are marked with uppercase roman numbers (I, II, III, IV, V, VI, and VII) which were introduced in Fig. 1. Compaction interfaces were generated during the construction process of ASSESS. The most pronounced of them are marked with lowercase roman numbers (i, ii, iii, iv, and v). In particular, the reflection of compaction interface (iv) close to the reflection of the groundwater table (1) are difficult to separate from reflections from material interfaces. Reflections from confining walls are most visible around $1\,\mathrm{m}$ (W) but influence the signal for more than $2\,\mathrm{m}$. The reflection of the edge of the L–element (L) is particularly prominent. As ASSESS is confined by walls at all four sides and approximately $4\,\mathrm{m}$ wide, the walls parallel to measurement direction also influence the measured signal.

The time-lapse GPR measurement was recorded at $17.05\,\mathrm{m}$ (Fig. 13b). As the groundwater table is raised, the reflection originating from the groundwater table (2) separates from the reflection of the compaction interface (iv) and its amplitude increases. After passing the material interface, the reflection (2) splits in two separate reflections (3) and (4) due to the strong dependency of the hydraulic conductivity on the water content (Sect. 3.2.1). As the transition zone is smoothing during the equilibration phase, the amplitude of reflection (3) decreases and the distance of the reflections (3) and (4) increases. During the subsequent imbibition step, the reflections are directly separated. According to the analysis of the synthetic data (Sect. 3.2.1), the effects of the smoothing water content distribution are most clearly visible during the relaxation phase at the reflections (5) and (6). However, the associated measured signals interfere with the direct wave, the ground wave, and the reflection from the compaction interface (i) which exacerbates the identification of these effects. The reflections (7) and (8) measured during the final



30 imbibition phase confirm the previous observations.

Together with the water content distribution, the time-lapse GPR data also contain information about the subsurface architecture. However, separating signal contribution from the subsurface architecture and the hydraulic dynamics is not always possible. Here, this is most prominent for the reflection of the material interface (V). Initially, the amplitude of this reflection is large because the water content in material C is near the residual water content, whereas the water content in material A is significantly higher at the material interface. As soon as both materials are water saturated, the amplitude of the material interface reflection (V) is low since the effective porosities of the two materials are similar. Thus, the amplitude of the reflected

5 signal originating from the material interfaces may change depending on the hydraulic state.

Additional information about the subsurface architecture can be inferred from the reflection at the material interface between material A and the gravel layer (VI) and from the reflection at the material interface of the gravel layer and the concrete basement (VII). These reflections are in particular suitable to analyze the total change of water content over time.

In summary, we note that the characteristic properties of the transition zone reflection during the imbibition and equilibration

10 steps that were identified in the simulation (Fig. 8) can also be identified in the measured data (Fig. 13).

### 3.3.2 Results and discussion

As the GPR measurements cover only a small portion of the subsurface architecture, we restricted the hydraulic representation to 1D (Sect. 3.1.1). Hence, we neglect 2D effects such as lateral flow. This has to be considered during the event selection of measured data (Sect. 2.3.3). Therefore, we merely focus the evaluation on the imbibition phase of the experiment as the effect of lateral flow in fluctuating groundwater table experiments is largest during drainage and close to the capillary fringe (Jaumann and Roth, 2017).

The identification of relevant events is more difficult for the measured data than for synthetic data due to additional reflections

of the GPR signal which are not represented. The most important representation errors are presumed to be the (i) reduced dimensionality of the representation of the inherently three dimensional GPR antennas and ASSESS test site using a 1D hydraulic and a 2.5D electrodynamic model, (ii) neglected small-scale heterogeneity in particular associated with compaction interfaces, (iii) neglected reflections from confining walls, (iv) neglected roughness of material interfaces, (v) influences of the antenna characteristics on the GPR signal in particular including the direct wave, the ground wave, temporal fluctuation of

time-zero, and the source wavelet, (vi) assumption of a constant direct current conductivity, and (vii) assumption of a constant soil temperature for the calculation of the relative permittivity of water.

The main findings concerning the mean parameters for the synthetic data (Sect. 3.2.2) can also be identified for the measured data, i.e. (i) the shift in the soil water characteristic of material C, (ii) the large uncertainty of the saturated hydraulic conductivity of material C, (iii) the high uncertainty of the soil water characteristic of material A for low water contents, and (iv) the

high sensitivity on $K_{s,A}$.

Compared to the uncertainties based on synthetic data (Table 2), the uncertainty of the resulting mean parameters (Table 3) mostly increased due to representation errors. Yet, the parameters estimated from TDR measurements (Jaumann and Roth, 2017) are within the standard deviation of the mean parameter set except for four parameters (Table 3). The deviations of these





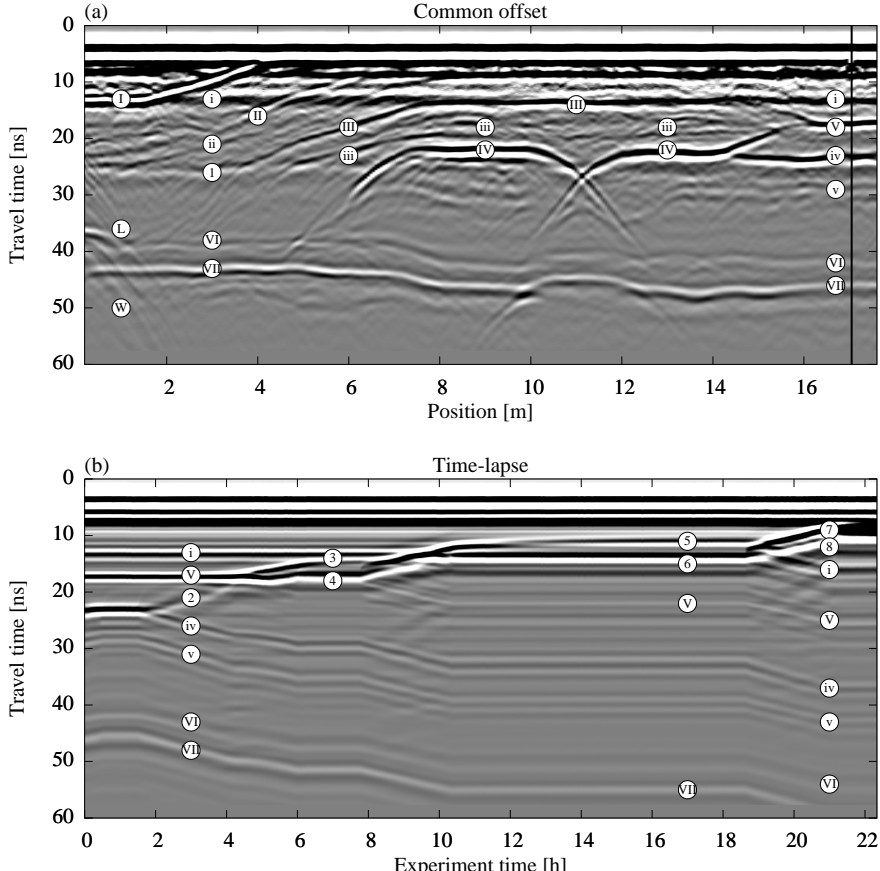

**Figure 13.** We show a common offset measurement of the hydraulic state of ASSESS at the beginning of the experiment (a). The vertical line indicates the position of the time-lapse measurement shown in subfigure (b). The common offset (time-lapse) data was measured with antenna 2 (3). Hence, in particular the measured GPR signal of the direct wave and the ground wave is slightly different. Both radargrams are measured with internal channels with an antenna separation of $a = 0.14$ m. Except for the normalization, the data are processed according to Sect. 2.3. In contrast to the quantitative evaluation, the radargram is normalized to the maximal absolute amplitude, facilitating the visual comparison of the traces. The markers are used consistently in this paper and are further explained in the text.





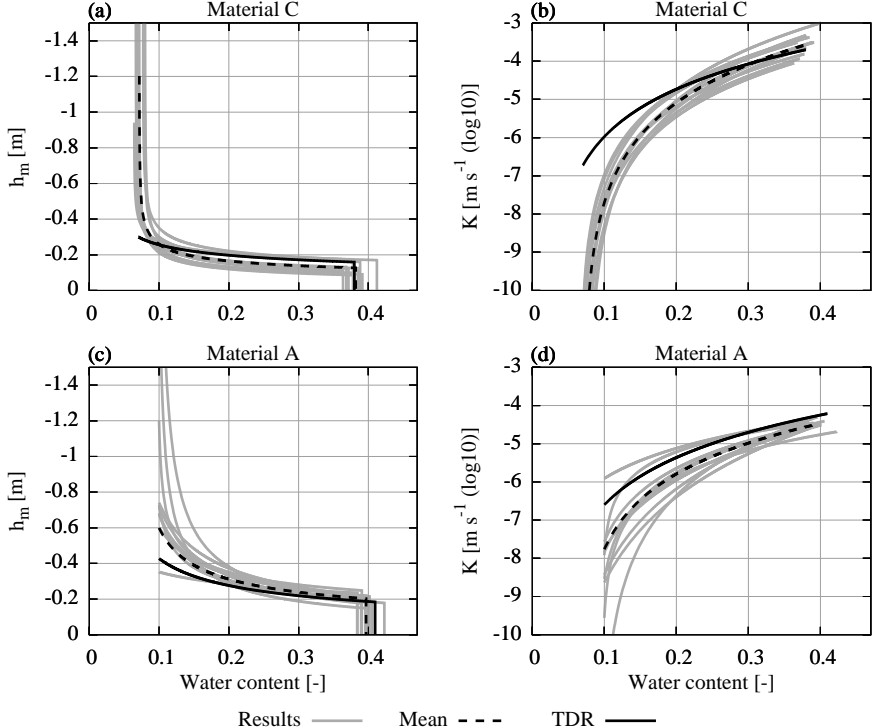

**Figure 14.** The resulting material parameters estimated from measured data are shown for the 10 best ensemble members (Sect. 3.1.2) together with the mean of these parameter sets (Table 3) and a reference parameter set determined from TDR data acquired during the same experiment (Jaumann and Roth, 2017). The plot range of the parameters is adjusted to the water content range of the corresponding data.

parameters are analyzed in the following.

The parameter $\theta_{r,C}$ estimated from the GPR data is approximately a factor 3 larger than the estimated value based on the TDR data. Essentially, there are three main reasons for this. First, by evaluating the travel time of reflection (V), we use the integrated water content for inversion. This also includes the compaction interface (i) which is not represented in the model. At the beginning of the experiment, the amplitude of this reflection is comparable to the amplitude of the reflection originating from the interface of material A and C (V). Notice that the amplitude of the reflection (i) does not vanish, but decreases when

the material is saturated at the end of the experiment. This indicates that this reflection originates from changes in both the small-scale texture of the material and the stored water content at the beginning of the experiment. Hence, as this compaction interface is not represented in the model, the resulting $\theta_{r,C}$ is increased coping for this representation error. Second, a deviation in the position of the groundwater table with reference to the antenna position at the surface can be partially adapted by changing $\theta_{r,C}$. As the position of the surface is subject to change over the years, the measurements of the groundwater table

are referenced to a fixed point at the end of the groundwater well, leaving the exact position of the surface relative to groundwater table uncertain. According to Buchner et al. (2012), the accuracy of the ASSESS architecture when compared to GPR





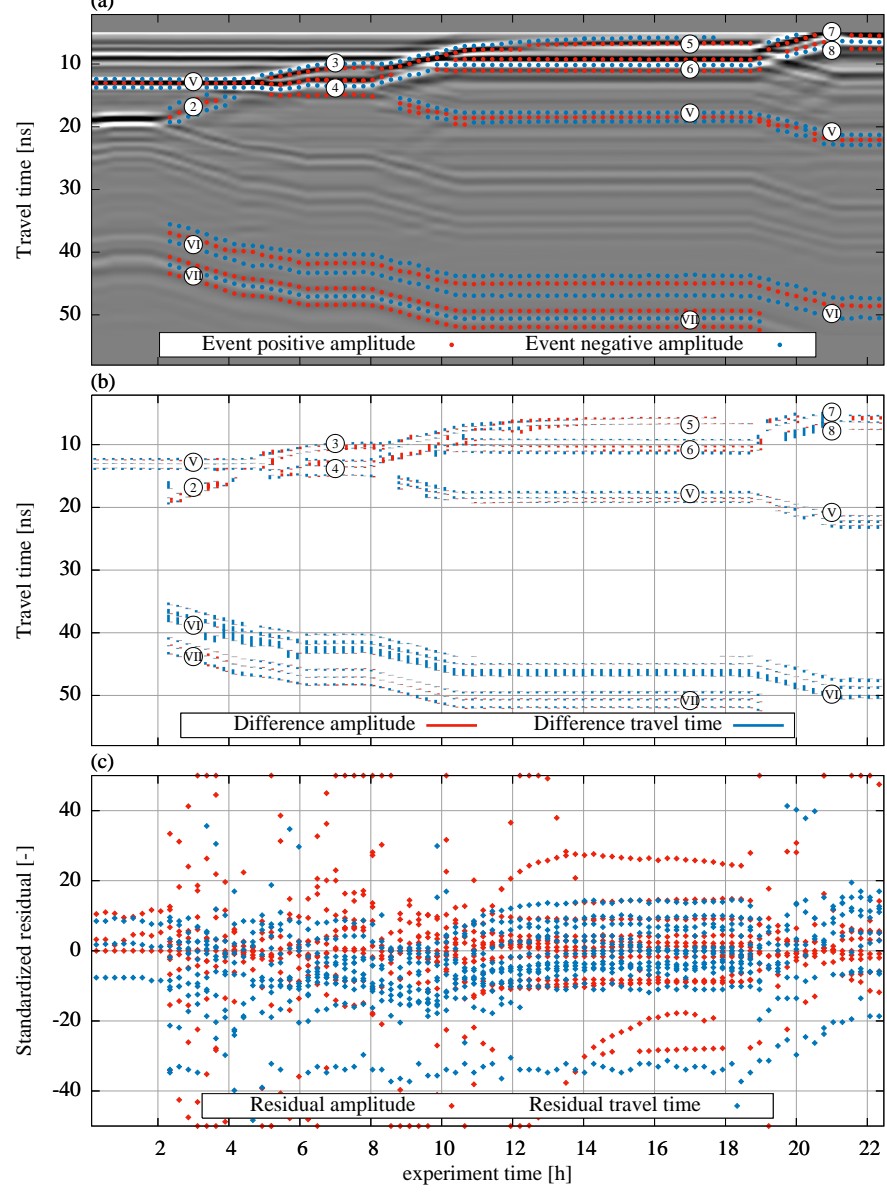

**Figure 15.** The evaluation the measured GPR data is separated in three parts: Subfigure (a) shows the detected (Sect. 2.3.2) and selected (Sect. 2.3.3) events which are used as synthetic measurement data. Except for the normalization, the data are processed according to Sect. 2.3. The radargram is normalized to the maximal absolute amplitude, facilitating visual comparison of the traces. Subfigure (b) shows resulting differences in travel time and amplitude of the mean parameter set. The differences of the amplitude are given in arbitrary units which are consistent for the whole work. Subfigure (c) shows standardized residuals of the differences given in subfigure (b). Note that outliers are set onto the boundary. The markers for the reflections are used consistently in this paper. Since the hydraulic 1D model cannot represent lateral flow present in initial drainage, we neglect the measured events of the first 2 h. Additionally, outlying events which have a different sign amplitude are not considered for event association (Sect. 2.3.4).





measurements is $\pm 0.05$ m. The estimation of an offset to the Dirichlet boundary could mitigate this problem, but would in any case increase the number of local minima significantly making the optimization process less stable. Third, evaluating the TDR data, we find that an underestimation of $\theta_{r,\mathrm{C}}$ is likely due to the lack of TDR measurements at low hydraulic potential. Hence,

the underestimation of $\theta_{r,\mathrm{C}}$ can be compensated with $h_0$ and $\lambda$.

Compared to the evaluation of the TDR data, the resulting value for parameter $\tau_{\mathrm{C}}$ is a factor 2 larger for the GPR evaluation. This parameter adjusts the hydraulic conductivity for the unsaturated material and is mainly determined with the reflections (3) and (5) originating from the additional kink during imbibition (Fig. 15). These reflections exhibit a small amplitude for low water contents. Yet, both reflections interfere with the rather prominent reflection originating from the compaction interface

(i). Additionally, the reflection (5) also interferes with parts of the direct and the ground wave. Hence, the travel time of these reflections does hardly change leading to an underestimation of the hydraulic conductivity for low water contents.

Similar to parameter $\theta_{r,\mathrm{C}}$, the parameter $\theta_{r,\mathrm{A}}$ yielded from the GPR evaluation is approximately a factor 3 smaller than the result from the TDR evaluation. However, this parameter can only be approximated evaluating the GPR data as they lack events that are influenced by low water content.

The resulting value for parameter $K_{\mathrm{s,A}}$ is factor 2 smaller for the GPR evaluation compared to the result from the TDR evaluation. This parameter limits the flux through the lower boundary as the domain is forced with a Dirichlet boundary condition. Hence, the parameter can be used to cope with errors in the boundary condition. Forcing ASSESS with a groundwater well instantiates a 3D flux (Jaumann and Roth, 2017). Since this is not represented, the hydraulic potential and hence the water flux is overestimated. This is compensated in the GPR evaluation by decreasing $K_{\mathrm{s,A}}$.

The estimated interface position of the material A and C corresponds well the to ground truth measurements acquired during the construction of ASSESS (Table 3). In contrast, the estimated position of the gravel layer deviates from the according ground truth measurements. However, the estimates are still within the uncertainty of the ground truth measurements when compared to GPR measurements.

The evaluation of the GPR measurement data (Fig. 15) shows that deviations from the shape of the reflected wavelet contributes

to the residuum significantly. These deviations have three main origins: (i) unknown shape of the source wavelet of the GPR antenna when coupled to the subsurface, (ii) the assumption of a constant direct current conductivity for the whole subsurface, and (iii) neglected roughness of the material interfaces which influences the shape of the reflected wavelet.

Evaluating the different reflections (V) and (VII), the residuals in travel time suggest an incorrect width of the wavelet. At the beginning of the experiment, the simulated wavelet is too broad for reflection (V) whereas it is too narrow for the reflection

(VII). This indicates that (i) the assumption of a constant direct electric conductivity in the whole architecture is suboptimal and that (ii) the direct electric conductivity can be assessed with GPR measurements.

The residuals for reflection (VI) are the major contribution to the cost function as different representation errors are combined. Of all the events in the wavelet, the events with the largest travel time exhibit the highest residuals. Due to the large grain size of the gravel, the real material interface is rougher than its representation. This leads to a non–symmetric broadening of the

measured compared to the simulated wavelet (Dagenbach et al., 2013). The residual of the other events in the wavelet can be attributed to a combination of representation errors associated with the boundary condition, suboptimal parameters, incorrect





direct current conductivity, and also to lateral water flow which not represented in 1D model.

A large part of the residuals associated to the reflections (3), (4), (5), and (6) originates from interferences with the compaction interface (i) which is not represented. Interferences typically are not quantitatively evaluable if not all contributions are cor-

rectly represented. This seems also to be the case for the signal during the forcing which is highly sensitive to the shape of the wavelet.

Regarding the total residuum, the error originating from assuming a constant soil temperature for the calculation of the relative permittivity of water is relatively small. However, it is worth noting, that the according residuals exceed one standard deviation in signal travel time.

Notice that the distribution and the support of the measurement data (i) differs between the TDR sensors and GPR measurements (Fig. 1), (ii) relates directly to the applicability of the resulting parameters for the different evaluations, and (iii) influences the quantitative effect of different representation errors. The TDR sensors are distributed over a 2D slice of AS-SESS measuring in all available materials (Fig. 1). Yet, the measurement volume is limited to the position of the sensors yielding the average permittivity along the TDR rods. Hence, these measurements are subject to representation errors such

as small-scale heterogeneity or uncertainty in the sensor position (Jaumann and Roth, 2017). The GPR measurement data do not cover the whole ASSESS test site and their support is depending on the evaluated events of the wavelet. This includes the whole depth average (travel time) and the contrast (amplitude) of both the permittivity and the electrical conductivity. Hence, these measurement data are subject to representation errors such as neglected (i) compaction interfaces, (ii) spatial variation of the direct current conductivity, and (iii) roughness of certain material interfaces. Hence, the previous analysis illustrates how

GPR-determined parameters can differ from TDR-determined ones making joint evaluation procedures challenging.

## 4   Conclusions

We demonstrated that effective soil hydraulic material properties and the subsurface architecture can be estimated accurately from single channel time-lapse GPR measurement data. The GPR-determined subsurface architecture corresponds well to the ground truth and the resulting material properties compare favorably to material properties determined from an independent

analysis of TDR measurements acquired during the same experiment.

These results are based on (i) a fluctuating groundwater table experiment at ASSESS, (ii) a new heuristic semi-automatic approach to automatically extract and associate relevant parts of simulated and measured radargrams, and (iii) an elaborate optimization procedure coupling different optimization algorithms which employ a subsampled data set to precondition the initial parameter proposals.

We confirmed that a fluctuating groundwater table experiment introduces characteristic transition zone reflections. A detailed qualitative analysis of its phenomenology indicates that this type of reflection is sensitive on soil hydraulic material properties. Employing the presented approaches on synthetic data shows that the true parameters are within the standard deviation corresponding to the resulting mean parameter set based on the ten best ensemble members. This mean parameter set describes the hydraulic dynamics with a mean absolute error in volumetric water content of 0.004. Additionally, we found that the parameter




correlations are mostly specific to the experiment type and the subsurface architecture. Using travel time and amplitude information in the evaluation allowed to estimate effective permittivity and layer depth simultaneously with a single GPR channel. The resulting parameters for the measured data are mostly consistent with results from the TDR measurement data. We discussed the deviations of the parameters and basically associated them with representation errors or the lack of available measurement data. Critical representation errors comprise the neglected (i) compaction interfaces, (ii) spatial variation of the direct current conductivity, and (iii) roughness of certain material interfaces.

## 5    Data availability

The underlying measurement data are available at http://ts.iup.uni-heidelberg.de/data/jaumann-roth-2017-gpr-hess.zip

*Author contributions.*    S. Jaumann designed and conducted the experiment, developed the main ideas, implemented the algorithms, and analyzed the measurement data. K. Roth contributed with guiding discussions. S. Jaumann prepared the manuscript with contributions of both authors.

*Acknowledgements.*    We thank Jens S. Buchner for the codes to process the ASSESS architecture raw data and the GPR data. We are grateful to Angelika Gassama for technical assistance with respect to ASSESS. We especially thank Patrick Klenk and Elwira Zur for assistance

during the experiment. The authors acknowledge support by the state of Baden-Württemberg through bwHPC and the German Research Foundation (DFG) through grants INST 35/1134-1 FUGG and RO 1080/12-1.



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





**Table 1.** The fit range limits the parameter space available for parameter estimation and is in particular used by the simulated annealing algorithm to draw parameter updates (Sect. 2.4.2). The sample range is used to generate an ensemble of initial parameter sets with the Latin hypercube algorithm.

| Material | Parameter | Fit range | | Sample range | |
|---|---|---|---|---|---|
| | | min | max | min | max |
| C | $h_0$ (m) | $-0.25$ | $-0.05$ | $-0.20$ | $-0.10$ |
| | $\lambda$ $(-)$ | 1.0 | 5.0 | 2.0 | 4.0 |
| | $K_0$ (m s$^{-1}$) | $10^{-4.1}$ | $10^{-2.9}$ | $10^{-4}$ | $10^{-3}$ |
| | $\tau$ $(-)$ | $-1.0$ | 2.0 | 0.0 | 1.0 |
| | $\theta_s$ $(-)$ | 0.33 | 0.43 | 0.36 | 0.40 |
| | $\theta_r$ $(-)$ | 0.00 | 0.10 | 0.02 | 0.08 |
| A | $h_0$ (m) | $-0.30$ | $-0.10$ | $-0.25$ | $-0.15$ |
| | $\lambda$ $(-)$ | 1.0 | 5.0 | 2.0 | 4.0 |
| | $K_0$ (m s$^{-1}$) | $10^{-5.1}$ | $10^{-3.9}$ | $10^{-5}$ | $10^{-4}$ |
| | $\tau$ $(-)$ | $-1.0$ | 2.0 | 0.0 | 1.0 |
| | $\theta_s$ $(-)$ | 0.36 | 0.46 | 0.39 | 0.43 |
| | $\theta_r$ $(-)$ | 0.00 | 0.10 | 0.02 | 0.08 |
| Gravel | $\theta_s$ $(-)$ | 0.30 | 0.50 | 0.38 | 0.42 |
| Architecture | $h_1$ (m) | 0.90 | 1.10 | 0.95 | 1.05 |
| | $h_2$ (m) | 0.10 | 0.30 | 0.15 | 0.25 |





**Table 2.** The mean and the standard deviation are calculated using the resulting parameters from the ten best ensemble members (Sect. 3.1.2) estimated from synthetic data. The corresponding material functions are given in Fig. 9. Notice that the true parameter set lies within the standard deviation of the mean parameter set.

| Material | Parameter | Truth | Mean results |
|---|---|---|---|
| C | $h_0$ (m) | $-0.15$ | $-0.13 \pm 0.02$ |
| | $\lambda$ $(-)$ | $3.5$ | $3.2 \pm 0.3$ |
| | $K_0$ (m s$^{-1}$) | $10^{-3.5}$ | $10^{-3.4 \pm 0.2}$ |
| | $\tau$ $(-)$ | $0.5$ | $0.6 \pm 0.2$ |
| | $\theta_s$ $(-)$ | $0.38$ | $0.38 \pm 0.01$ |
| | $\theta_r$ $(-)$ | $0.03$ | $0.027 \pm 0.006$ |
| A | $h_0$ (m) | $-0.20$ | $-0.199 \pm 0.008$ |
| | $\lambda$ $(-)$ | $2.5$ | $2.8 \pm 0.7$ |
| | $K_0$ (m s$^{-1}$) | $10^{-4.5}$ | $10^{-4.47 \pm 0.05}$ |
| | $\tau$ $(-)$ | $0.5$ | $0.4 \pm 0.5$ |
| | $\theta_s$ $(-)$ | $0.41$ | $0.41 \pm 0.02$ |
| | $\theta_r$ $(-)$ | $0.05$ | $0.06 \pm 0.02$ |
| Gravel | $\theta_s$ $(-)$ | $0.40$ | $0.40 \pm 0.03$ |
| Architecture | $h_1$ (m) | $1.00$ | $0.99 \pm 0.02$ |
| | $h_2$ (m) | $0.20$ | $0.20 \pm 0.01$ |





**Table 3.** The mean and the standard deviation are calculated using the resulting parameters from the ten best ensemble members (Sect. 3.1.2) estimated from measured data. The corresponding material functions are given in Fig. 14. The reference parameters for the materials A and C are determined from TDR data acquired during the same experiment investigated in this work (Jaumann and Roth, 2017). Note that the according standard deviations are determined formally from a single Levenberg–Marquardt run and hence are only representative for one local minimum. Also, these standard deviations are given with the understanding that they are specific to the applied algorithm and will change for different algorithm parameters. Hence, these standard deviations are in particular not suitable to compare the precision of the TDR and GPR evaluation. Notice that for the TDR evaluation the porosity of the materials is assumed to be known from core samples. The reference parameters for the subsurface architecture are calculated from ground truth measurements acquired during the construction of ASSESS. The corresponding standard deviations are given according to Buchner et al. (2012).

| Material | Parameter | Reference | Mean results |
|---|---|---|---|
| C | $h_0$ (m) | $-0.159 \pm 0.004$ | $-0.13 \pm 0.03$ |
|  | $\lambda$ (−) | $3.28 \pm 0.02$ | $3.3 \pm 0.7$ |
|  | $K_0$ (m s$^{-1}$) | $10^{-3.70 \pm 0.02}$ | $10^{-3.6 \pm 0.3}$ |
|  | $\tau$ (−) | $0.74 \pm 0.06$ | $1.4 \pm 0.4$ |
|  | $\theta_s$ (−) | $0.38$ | $0.38 \pm 0.01$ |
|  | $\theta_r$ (−) | $0.026 \pm 0.002$ | $0.071 \pm 0.005$ |
| A | $h_0$ (m) | $-0.184 \pm 0.005$ | $-0.20 \pm 0.03$ |
|  | $\lambda$ (−) | $1.94 \pm 0.07$ | $2.1 \pm 0.7$ |
|  | $K_0$ (m s$^{-1}$) | $10^{-4.212 \pm 0.004}$ | $10^{-4.5 \pm 0.1}$ |
|  | $\tau$ (−) | $0.33 \pm 0.07$ | $0.4 \pm 1.0$ |
|  | $\theta_s$ (−) | $0.41$ | $0.40 \pm 0.01$ |
|  | $\theta_r$ (−) | $0.025 \pm 0.004$ | $0.07 \pm 0.03$ |
| Gravel | $\theta_s$ (−) |  | $0.41 \pm 0.02$ |
| Architecture | $h_1$ (m) | $0.99 \pm 0.05$ | $0.97 \pm 0.02$ |
|  | $h_2$ (m) | $0.13 \pm 0.05$ | $0.17 \pm 0.02$ |