# Peer review of "Soil hydraulic material properties and layered architecture from time-lapse GPR"

_Hydrology and Earth System Sciences, 2017_

## Referee Comment (RC1) · Anonymous Referee #1 · 15 Sep 2017

Jaumann and Roth consider the problem of inferring for soil hydraulic properties for the situation when (1) hydraulic and electrical properties only vary with depth; (2) interface locations are known and they are horizontal; (3) the initial position of the water table is known. The data at hand are GPR traces acquired over time during a controlled variation of the water table. The experiments are carried out at the ASSESS test site, a large-scale facility for studying the use of GPR data in vadose zone hydrology with known layering, control of water table and supplementary data, such as, time-domain reflectometry data. The inversion methodology uses a 1-D solver to the Richards equation, a 2-D electromagnetic forward, and a rather elaborate (but also somewhat convoluted) optimization algorithm with little demonstration that it enables

the localisation of global minima and that the ensemble members used provide a reasonable estimate about parameter uncertainty. One of the key aspects of this algorithm is the automatic detection of events (typically negative and positive peaks) in the GPR traces. The writing is overall good, but I also feel that the manuscript could be simplified and shortened.

1. The authors state in title and abstract that they estimate the subsurface architecture. Reading this, I expected the retrieval of 2-D or 3-D geometry of lithofacies. Instead, the authors assume a known layered system and "simply" infer for parameters in the Mualem model (and allow for some very small variations in interface locations). I find this terminology to be inappropriate, and it should be stated that the authors infer for hydraulic properties of multiple known layers. From what I understand, this is the main novelty of this work: using water table fluctuations to infer hydraulic material properties for more than one layer (2 in this case; much less significant than the statement on line 5 in abstract).

2. The authors need to put this work into context. How does the presented work improve understanding about vadose zone processes, how can the used method be used for actual field applications (not that easy given that it is assumed that everything is 1-D and that interfaces are known, which is seldom the case)? Many readers are likely to question why to go through all this trouble instead of doing the same inversion using a few TDR probes. The answer is related to larger-scale applications, but this is not handled here (only one GPR position). A clear motivation is needed in introduction and before the conclusions. In short, why should someone that is not working at ASSESS read this work and how can it advance hydrology or the use of GPR to characterise hydrology. This is not clear reading the present version of the manuscript.

3. It is disappointing to only see applications in 1-D. What is the reason for not modeling flow in 2-D, to use all three monitoring locations, and the full extent of the water table fluctuations? Is this something you plan to do in the future? Also, how to deal with the fact that the 1-D representation is unsuitable for significant drainage? After seeing

Figure 1 and 2, it is easy to be a bit disappointed when only seeing results that consider GPR position 3 and no significant drainage. The tank has a nice 2-D subsurface architecture, but it is here simplified to a known 1-D layered system.

4. Figure 1: Make it very clear in figure and caption that you only consider data from GPR antenna 3. It is somewhat confusing to see this spatial representation, while all the treatment relates to one GPR position. I would not use the term "radargram" to represent time-series of the GPR traces, as radargrams (e.g., page 6, line 7) are often thought of as a time-distance plot. Make it clear in the text that all the GPR results and simulations only model the trace at a given location over time and that no spatial information is treated (except depth).

5. Explain clearly what is meant by subscale physics. These are all macroscopic representations, so why call them "subscale".

6. I recommend that some pseudo-code is added for the algorithm used in 2.3.4. Is the method practical for 2-D and 3-D applications?

7. 2.4 is called parameter estimation, but it is never written that the parameterization in terms of geology is assumed to be known +- epsilon. This is a very strong assumption and it would be much more difficult to solve the inverse problem if one would actually infer the "subsurface architecture".

8. Equation 11. How are the standard deviations estimated in practice (see also page 15, line 11)? Are they due to observational errors (estimated how), modeling errors (estimated how) or purely ad hoc? Page 3, line 9: What is the implication of excluding data events for the global optimizer (simulated annealing) used?

9, lines 5-6: I don't understand this statement at all. Is this simply related to the fact that you damp the update size or is it something else?

10, page 13, line 5: Why not estimate the source wavelet as a part of the inversion (frequency and shape)?

11. A transition is really needed when starting 3.2.1. Write explicitly that you first will consider a synthetic test case to gain knowledge about the information content in the data and the ability of the inversion to provide a reasonable model. Explain the geometry of this model, explain how noise was added to the generated data. Similarly, a transition is needed when starting 3.2.2 (e.g., After inversion, we find that the...)". In Figure 9, add estimated "by inversion".

12. Work a bit on the definitions of paragraphs (e.g., page 26).

13. Reconsider the use of phenomenology in favour of more common language in hydrology:

"the science of phenomena as distinct from that of the nature of being. • an approach that concentrates on the study of consciousness and the objects of direct experience."

Smaller comments (suggestions):

Page 1, line 2: Should be "Ground..." not "ground..." Replace "to" with "that is suitable to" to clarify that the GPR method was not built explicitly for this application. There are many more applications of GPR.

Page 1, line 3: Remove "precisely". It is clear that a quantitative method (pretty clear) is used, so what is precisely supposed to mean? Especially given the rather low agreement with TDR estimates for the field data.

Page 1, line 8: Perhaps explain what an "association algorithm" is.

Page 1, line 20: Replace "monitors the hydraulic processes accurately" to "is sensitive to hydraulic processes". What is monitored with a TDR is essentially the dielectric constant, which indirectly is related to hydraulic processes.

Page 2, line 6: Remove "are the easiest and" with "offer". Maybe the measurement procedure is easier, but the real work is in the modeling and inversion. Not clear to me

why this mode is easier than say borehole data.

Page 2, line 12: Add "indirect" before "information".

Page 2, line 14: add "to reproduce when used" before "for".

Page 2, line 15: Replace "in" with "for", remove ")". Sentence starting on line 15 is not clear. Why is this information not as important when considering precipitation or flooding events?

Page 2, line 23: Remove "quantitatively", this statement does not add anything.

Page 2, line 23: Remove "balance" with "are faced by an inherent trade-off between"

Page 2, line 27: A fair bit of self-referencing throughout. Why not cite some of the many other works related to GPR modeling.

Page 3, line 7: Replace "may even" with "lead to better convergence and may even"

Page 3, line 16 (and many other places. It should be "sensitive to", not "sensitive on".

Page 3, line 19: I know that both uses are correct, but I prefer to treat data in plural form: One datum, several data.

Page 3, line 12: Add "for porous media" before "the standard".

Page 4, line 18: Explain already here that the reason for ignoring the large drainage event is that Richards equation is solved in 1-D.

Page 6, line 8: Why only one antenna? Why not model this as a 2-D system?

Page 6, line 15: Why conductivity at dc conditions. It should be the conductivity at around 400 MHz, typically 50% or so higher than the DC value.

Page 6, line 20: Clarify that one normally has no idea about the power of the GPR source, only some basic idea about its shape. This implies that some sort of normalization of observed and simulated traces are needed.

Page 6, equation 7. Here, the dependence of temperature is included, but it is written later that this was not done and it led to errors on the scale of one standard deviation in time. Also, how are boundary conditions in the tank modeled in the EM code?

Page 6, line 31: Use Archie to explain how big this approximation is. Depends on the differences in porosity of the sands used.

Page 7, line 4: Should be "corresponding" instead of "corresponding".

Page 7, lines 13-14: Confusing as treatment to simulated and observed data are mixed. For example, (ii) only important for real data and (iii) only needed for simulated data. Please clarify what is done for (1) simulated data and (2) observed data.

Page 7, line 18: Is this correction valid for a dipole radiation pattern.

Page 7, line 23: "i" in italics.

Page 27, line 19: Replace "favourably" with "reasonably well".

―――――――――――――――

---

## Referee Comment (RC2) · Anonymous Referee #2 · 16 Nov 2017

The authors deduce subsurface hydraulic properties by an inversion of time-lapse surface GPR measurements during an imbibition experiment of an artificial test site. The coupled inversion process includes a hydraulic simulation by solving the 1D Richards equation and a simulation of radar wave propagation by 2D finite-differences calculation. Water content distribution and electromagnetic soil properties are coupled by a petrophysical relation (CRIM). During the inversion, the misfit between events, i.e. traveltimes and amplitudes of selected reflections, in experimental and synthetic GPR data is minimised. The authors use an inversion scheme that combines several optimisation steps including global and gradient techniques. The approach is first demonstrated for synthetic data and later for experimental data. The result is a 1D subsurface model and

for both predefined layers the characteristic hydraulic properties of a Brooks-Corey parameterisation of the water retention function is fitted.

The presented work is a relevant contribution towards a non-destructive hydraulic characterisation of the subsurface, which is still an unsolved problem for the unsaturated zone. However, the manuscript has to be overworked as the whole analysis including GPR data processing and inversion is somehow nebulous and difficult to follow. I would also suggest to shorten the text by writing more tersely, avoiding repetitions and possibly moving some parts into an appendix as e.g. GPR data conversion due to Bleistein, details on event detection/association and inversion. Besides this, some major points have to be clarified:

1. Amplitude handling: The formula used for spherical divergence correction for 3D data (P 7 Eq. 10) seems not correct. Correcting with square root of distance is used for 2D data. Also the dimensions of Eq. 10 do not fit. Various formulations of adequate gain functions for 3D (experimental) data is given in Yilmaz: Seismic Data Analysis (2001), e.g. Eq. (1-8a) $g(t) = \frac{v^2(t)t}{v_0^2 t_0}$. The whole amplitude balancing in the manuscript is not clear to me. The radar traces are normalised several times (P8 L1-2) and normalisation is done relative to the maximal absolute amplitude, which is the first reflection. Is the reflector characteristics constant during the entire experiment? What is the advantage of the complicated amplitude adaption due to Bleistein (1986) compared to a simple correction of 2D circular divergence with the square root of the distance? I would suggest to provide a flow chart of amplitude handling for both experimental (3D) and synthetic (2D) radar data. It seems you use different amplitude handling for event detection and the inversion process?

2. Neglecting dielectric losses. I'm wondering if at frequencies of about 400 MHz, the impact of free water relaxation can be neglected and whether the imaginary part of permittivity has to be taken into account. When using complex permittivity

of water according to Kaatze et al. (1989) and the CRIM formula and a DC conductivity of 0.003 S/m this results in: 3 dB/m (2 dB/m from free water relaxation, 1 dB/m from DC conductivity) for 10vol% water content and 5 dB/m (4 dB/m from free water relaxation and 1 dB/m from DC conductivity) for full saturation (40vol% water content). This means that up to 80% of total loss is caused by polarisation effects of free water. Neglecting these effects results in wrong amplitudes of the simulated data and I'm wondering how they can fit to the field data. At the end of the imbibition experiment (water table at -0.6 m) the amplitudes of the lower reflections (1 m saturated material above, i.e. 2 m two-way travel path) should appear to be approximately 8 dB (2.5 times) higher in the synthetic data than in the field data. I suggest to use the true complex permittivity of water at 400 MHz or, if the FDTD code cannot handle complex property values, an effective HF conductivity including both, DC conduction losses and HF polarisation losses.

3. GPR forward calculation: Why is a 2D FDTD code used for a horizontally layered model? A 1D reflectivity method as e.g. used by Bradford et al., (2014) or a 1D FDTD code would be much more efficient. The power of FDTD is certainly that it can be used for complicated 2D/3D subsurface models and thus for inverting 2D/3D data with an according hydraulic simulation. However, in the presented study only 1D data are used and no outlook is given how to adopt the strategy to 2D or 3D problems. From this it is not clear why the expensive 2D FDTD algorithm is used. The source wavelet of the simulation is different to the wavelet of the experimental data. When dealing with gradient interfaces as the capillary transition zone, the wavelet shape may have a big impact on the maximal amplitude of the reflected signal. Why not using the first reflected signal, which is used for normalisation, as source wavelet in the simulation?

4. Inversion. The complex inversion scheme is a nesting of global and gradient methods. It is somewhat nebulous and it's difficult to get an impression of the quality of original data fit. Why do you use different but relatively narrow boundaries (fit ranges) for the inversion parameters of the two layers? By doing this, the inversion result is biased by a-priori information that is usually not known but the actual aim of the investigation. The inversion should work with the same (broader) fit range for both layers. If not, it cannot be adopted to the field. The fit ranges should be used to provide outer boundaries of the deduced material properties in Fig. 9 and Fig. 14. I'm also missing a figure showing experimental GPR data traces and synthetic traces based on the inversion models to prove that the experimental data are well described. This figure should include the resulting synthetic radar traces of the ten best inversion results to get an idea of the fluctuations and an idea of the fitting quality.

5. In the analysis, the reflection of the compaction layer is excluded. If this interface causes a GPR reflection, this must be caused by different water contents on both sides and hence, there must be significant differences in the material hydraulic properties (see P24 L25ff). So why should I ignore an interface that is present in the subsurface and reflects changes in hydraulic properties? Please explain.

6. The title is misleading, I would suggest to delete "...and subsurface architecture..." as this would imply at least a 2D subsurface model. The section headings of 2.2 sound unusual to me. From a geophysical perspective the following headings would give a better description: 2.2.1 Water dynamics, 2.2.2 Hydraulic material characterisation 2.2.3 Time lapse experiment 2.2.4 GPR investigation and electromagnetic material characterisation.

Further comments

- (P2 L27) References are a bit biased by the own workgroup. E.g., when introducing the FDTD method I would expect the basic work of Yee, Taflove... and,e.g., the former ETHZ geophysics group or from the gprMax developers.

- (P6 L23ff) The CRIM formula uses the square root of permittivities (see your original reference: Birchak et al., 1974). There is no need to first define a general formulation with an exponent $\alpha$, which is not the original CRIM formula, and then fix the exponent $\alpha = 0.5$. Keep it simple and use the square root from the beginning.

- (P6 L9) You should describe that you use the static permittivity of water (which is acceptable for 400 MHz, at least for the real part of permittivity).

- (P7 L14) "...removal of the direct and trailing signal". What is the trailing signal? Is this the interference of ground wave, crosstalk, reflection at the ground surface and the antenna metal shielding? In Fig. 3, a part of this trailing signal is remaining, which is confusing. Why not muting this part?

- (P7 L15) "...we pick the direct signal and subtract it from the radargram" is confusing. Not the signal is subtracted but the travel time.

- (P8 L5) "normalized amplitude (original amplitude)". Rephrase, as the amplitude is either normalized or original.

- (P8 L6) "amplitude is amplified quadratically with travel time" means they are corrected for spherical divergence twice consecutively? Is this just an arbitrary gain function that showed to work well and to correct for spherical and intrinsic attenuation at the specific site? Please explain.

- (P11 L3) Eq. 11: I think the expression has to be divided by M to get the classical $\chi$ with $\chi^2 = 1$ if the data are described within the error.

- (P11 L4) How is the standard deviation of the normalised travel times and amplitudes calculated, i.e. what are the input data?

- (P13 L33) "...infinite dipole pointing in x dimension". This should be y dimension (into the plane of projection). Please provide x,y direction in Fig. 6.

- (P13 L34) a Ricker function is the second derivative of a Gauss-fct, not the first derivative

- (P15 L11) Please specify how you calculate the STD of the measured data. Do you really use all amplitudes of the radar trace and all travel times?

- (P18 L26) What is the meaning of amplitude information of a single channel? You are using a single channel GPR system and only one antenna, so this expression is confusing.

- (P24 L30) Couldn't the uncertainty of the groundwater table relative to the ground surface be overcome by simple levelling the ground surface?

- (P26 L20ff) is a partial repetition of (P22 L5ff (the lower line 5)).

- (P26 L26) "...and that (ii) the direct electric conductivity can be assessed with GPR measurements". I cannot understand the context.

- (P27 L13ff) This is again a partial repetition of (P22 L5ff) and (P26 L20ff).

- (P27 L18) Better use "constant offset" (CO) instead of "single-channel" GPR data.

- Fig. 3, caption. Are these synthetic or experimental data?

- Fig. 6: Which E-field component is shown, what is the x and y direction?

- Fig. 12, bottom: y-axis label: standardized residual: Does 10 mean that the residual is 10 times the STD or should it be 10

- Fig. 13: I suggest to split the figure into two individual figures as it might be very confusing to mix the 2D radar section with the time lapse data at one location. The label for the groundwater table reflection is "l" in the upper radar section and "2" in the lower time-lapse data. Actually, it's very hard to distinguish the label "l"

from the label "1" in the upper radar section. Please use the same and distinct labels for the GWT reflection in all figures.

**References**

Yilmaz, Ö.: Seismic Data Analysis, SEG, 2001.

Bleistein, N.: Two-and-one-half dimensional in-plane wave propagation, Geophysical Prospecting, 34, 686–703, doi:10.1111/j.1365-2478.1986.tb00488.x, 1986.

Kaatze, U.: Complex permittivity of water as a function of frequency and temperature, Journal of Chemical and Engineering Data, 34, 371–374, doi:10.1021/je00058a001, 1989.

Bradford, J., Thoma, M., and Barrash, W.: Estimating hydrologic parameters from water table dynamics using coupled hydrologic and 20 ground-penetrating radar inversion, in: Proceedings of the 15th International conference on Ground Penetrating Radar, pp. 232–237, doi:10.1109/ICGPR.2014.6970420, 2014.

Taflove, A. and Hagness, S.C.: Computational Electrodynamics: The Finite-Difference Time-Domain Method, Artech: Norwood, MA, 2005.

Birchak, J. R., Gardner, C. G., Hipp, J. E., 15 and Victor, J.M.: High dielectric constant microwave probes for sensing soil moisture, Proceedings of the IEEE, 62, 93–98, doi:10.1109/PROC.1974.9388, 1974.

---

## Author Comment (AC5) · 22 Dec 2017

Since the replies to the comments of the referees refer to a revised version of the manuscript, this version is attached to this author comment.

Please also note the supplement to this comment:
https://www.hydrol-earth-syst-sci-discuss.net/hess-2017-538/hess-2017-538-AC5-supplement.pdf

---

## Author Response (AR1)

**Reply to referee comment 1**

*Jaumann and Roth consider the problem of inferring for soil hydraulic properties for the situation when (1) hydraulic and electrical properties only vary with depth; (2) interface locations are known and they are horizontal; (3) the initial position of the water table is known. The data at hand are GPR traces acquired over time during a controlled variation of the water table. The experiments are carried out at the ASSESS test site, a large-scale facility for studying the use of GPR data in vadose zone hydrology with known layering, control of water table and supplementary data, such as, time-domain reflectometry data. The inversion methodology uses a 1-D solver to the Richards equation, a 2-D electromagnetic forward, and a rather elaborate (but also somewhat convoluted) optimization algorithm with little demonstration that it enables the localisation of global minima and that the ensemble members used provide a reasonable estimate about parameter uncertainty. One of the key aspects of this algorithm is the automatic detection of events (typically negative and positive peaks) in the GPR traces. The writing is overall good, but I also feel that the manuscript could be simplified and shortened.*

**Reply:** We thank the reviewer for the constructive comments and suggestions. We revised the manuscript accordingly and refer to the revised version in the following.

*1. The authors state in title and abstract that they estimate the subsurface architecture. Reading this, I expected the retrieval of 2-D or 3-D geometry of lithofacies. Instead, the authors assume a known layered system and "simply" infer for parameters in the Mualem model (and allow for some very small variations in interface locations). I find this terminology to be inappropriate, and it should be stated that the authors infer for hydraulic properties of multiple known layers. From what I understand, this is the main novelty of this work: using water table fluctuations to infer hydraulic material properties for more than one layer (2 in this case; much less significant than the statement on line 5 in abstract).*

**Reply:**

Previous work showed that the location of moderately complicated layer interfaces and of the mean water content between them can be obtained from multi-channel common offset measurements (Buchner et al., 2012). Together with the demonstration in this paper that the effective hydraulic material properties of layers can be estimated from single-channel time lapse measurements, we now have the methods to determine the subsurface architecture and its hydraulic properties for moderately complicated situations.

This obviously demands quite a significant experimental effort together with subsequent massive computations as time-lapse common offset measurements of the region of interest are required, which then have to be inverted.

The main novelty of this paper is the developed evaluation method that (i) is conceptionally not limited to layers but allows to evaluate moderately complicated subsurface architectures, (ii) does not require to estimate the full wave form but focuses on specific reflections chosen by the user, and (iii) can extract the information provided by changing shapes of the reflected wavelet. Thus, although the approach is demonstrated for a layered subsurface architecture, it is not limited to 1D or to simple layers.

*2. The authors need to put this work into context. How does the presented work improve understanding about vadose zone processes, how can the used method be used for actual field applications (not that easy given that it is assumed that everything is 1-D and that interfaces are known, which is seldom the case)? Many readers are likely to question why to go through all this trouble instead of doing the same inversion using a few TDR probes. The answer is related to larger-scale applications, but this is not handled here (only one GPR position). A clear motivation is needed in introduction and before the conclusions. In short, why should someone that is not working at ASSESS read this work and how can it advance hydrology or the use of GPR to characterise hydrology. This is not clear reading the present version of the manuscript.*

**Reply:**
We revised the manuscript accordingly (P1 L23ff, P3 L23ff, P29 L21ff, P30 L27ff, P30 L30ff).

*3. It is disappointing to only see applications in 1-D. What is the reason for not modeling flow in 2-D, to use all three monitoring locations, and the full extent of the water table fluctuations? Is this something you plan to do in the future? Also, how to deal with the fact that the 1-D representation is unsuitable for significant drainage? After seeing Figure 1 and 2, it is easy to be a bit disappointed when only seeing results that consider GPR position 3 and no significant drainage. The tank has a nice 2-D subsurface architecture, but it is here simplified to a known 1-D layered system.*

**Reply:**
As mentioned in the reply to remark 1, running the 2D or even 3D measurements and inversions is a massive experimental and computational effort. Prior to embarking on this, the individual steps must be demonstrated. This is the aim of the current paper, in conjunction with the earlier work of Buchner et al. (2012) that supplies all the prerequisites needed here. As the methods demonstrated here are capable of analyzing a number of measured radargrams simultaneously, they extend naturally to time-lapse common offset measurements acquired for more complicated subsurface architectures or during drainage conditions.

*4. Figure 1: Make it very clear in figure and caption that you only consider data from GPR antenna 3. It is somewhat confusing to see this spatial representation, while all the treatment relates to one GPR position. I would not use the term "radargram" to*

*represent time-series of the GPR traces, as radargrams (e.g., page 6, line 7) are often thought of as a time-distance plot. Make it clear in the text that all the GPR results and simulations only model the trace at a given location over time and that no spatial information is treated (except depth).*

**Reply:**

We agree and revised the manuscript accordingly (Fig. 1, Fig. 2, caption of Fig. 14). However, we did not separate the radargrams, because especially for people that are not used to time-lapse radargrams, having a corresponding common offset radargram of the initial state helps to associate the reflections and to understand their temporal evolution.

*5. Explain clearly what is meant by subscale physics. These are all macroscopic representations, so why call them "subscale".*

**Reply:**

The dynamics of the system is represented with a physics-based mathematical description for a predefined scale in space and time. In contrast, the physics below these scales is not represented explicitly. Instead, the macroscopic effects of this sub-scale physics are typically described heuristically. We clarified the manuscript (P4 L19ff).

*6. I recommend that some pseudo-code is added for the algorithm used in 2.3.4. Is the method practical for 2-D and 3-D applications?*

**Reply:**

We added a flowchart to explain the algorithm further (Fig. 3).

Going to 2D even to 3D is first and foremost a matter of computational effort with already 2D demanding significant time on a large computer cluster. No concepts or methods beyond of what we demonstrated in this paper are required, however.

*7. 2.4 is called parameter estimation, but it is never written that the parameterization in terms of geology is assumed to be known +- epsilon. This is a very strong assumption and it would be much more difficult to solve the inverse problem if one would actually infer the "subsurface architecture".*

**Reply:**

We agree but would like to point out, that estimating the architecture as demonstrated by Buchner et al. (2012) is a step that can (and should) be done prior to the actual estimation of the parameters because the architecture is invariant during the hydraulic experiment. Fine-tuning the position, as is done here, then ascertains that the entire inversion is self-consistent.

*8. Equation 11. How are the standard deviations estimated in practice (see also page 15, line 11)? Are they due to observational errors (estimated how), modeling errors (estimated how) or purely ad hoc? Page 3, line 9: What is the implication of excluding data events for the global optimizer (simulated annealing) used?*

**Reply:**

We clarified the manuscript (P17 L6ff).

The simulated annealing algorithm, as described in Sect. 2.4.2 is no global optimizer,

since the parameter update is only drawn from the neighborhood instead of the full parameter space. As the data set for low resolution is merely used for preconditioning, there are no significant implications. We also ran the inversions using random traces for preconditioning. This did not lead to significant changes in the results.

*9, lines 5-6: I don't understand this statement at all. Is this simply related to the fact that you damp the update size or is it something else?*

**Reply:**

We assume that the comment refers to page 11. The difference of the signal travel time and amplitude of associated events enters the cost function. Hence, if the number of associated events changes during the optimization, then the cost function becomes discontinuous. This happens, e.g., if the porosity of two saturated materials are similar during the optimization process. Then, the associated reflection will vanish and the measured events can not be associated anymore.

*10, page 13, line 5: Why not estimate the source wavelet as a part of the inversion (frequency and shape)?*

**Reply:**

We agree that the source wavelet does influence the reflected signal. Additionally, Dagenbach et. al. (2013), for example, showed that roughness of the material interfaces also influences the shape of the reflected wavelet. In particular to investigate the necessity to address these higher order uncertainties in a quantitative analysis, we did not represent them and investigated the structural residuals after the inversion. We propose that the effect of relevant representation errors on the estimated properties should be analyzed in a next step, similar as has been done for TDR data by Jaumann and Roth (2017). This is a significant effort well beyond the scope of this paper.

*11. A transition is really needed when starting 3.2.1. Write explicitly that you first will consider a synthetic test case to gain knowledge about the information content in the data and the ability of the inversion to provide a reasonable model. Explain the geometry of this model, explain how noise was added to the generated data. Similarly, a transition is needed when starting 3.2.2 (e.g., After inversion, we find that the. . .)". In Figure 9, add estimated "by inversion".*

**Reply:**

We agree and added transitions (P17 L12ff, P19 L17, and P23 L20ff).

*12. Work a bit on the definitions of paragraphs (e.g., page 26).*

**Reply:**

We revised the section 3.3.2 accordingly.

*13. Reconsider the use of phenomenology in favour of more common language in hydrology: "the science of phenomena as distinct from that of the nature of being. An approach that concentrates on the study of consciousness and the objects of direct experience."*

**Reply:**

We agree that this is one definition. However, the word is also used commonly in a range

of natural sciences from particle physics to meteorology (`https://en.wikipedia.org/wiki/Phenomenology`).

Smaller comments (suggestions):

*Page 1, line 2: Should be "Ground. . ." not "ground. . ." Replace "to" with "that is suitable to" to clarify that the GPR method was not built explicitly for this application. There are many more applications of GPR.*
*Page 1, line 3: Remove "precisely". It is clear that a quantitative method (pretty clear) is used, so what is precisely supposed to mean? Especially given the rather low agreement with TDR estimates for the field data.*
*Page 1, line 8: Perhaps explain what an "association algorithm" is.*
*Page 1, line 20: Replace "monitors the hydraulic processes accurately" to "is sensitive to hydraulic processes". What is monitored with a TDR is essentially the dielectric constant, which indirectly is related to hydraulic processes.*
*Page 2, line 6: Remove "are the easiest and" with "offer". Maybe the measurement procedure is easier, but the real work is in the modeling and inversion. Not clear to me why this mode is easier than say borehole data.*
*Page 2, line 12: Add "indirect" before "information".*
*Page 2, line 14: add "to reproduce when used" before "for".*
**Reply:**
We revised the manuscript accordingly.

*Page 2, line 15: Replace "in" with "for", remove ")". Sentence starting on line 15 is not clear. Why is this information not as important when considering precipitation or flooding events?*
**Reply:**
We revised the manuscript accordingly.
Generally, precipitation is spatially more homogeneously distributed compared to artificial irrigation.

*Page 2, line 23: Remove "quantitatively", this statement does not add anything. Page 2, line 23: Remove "balance" with "are faced by an inherent trade-off between"*
**Reply:**
We revised the manuscript accordingly.

*Page 2, line 27: A fair bit of self-referencing throughout. Why not cite some of the many other works related to GPR modeling.*
**Reply:**
Instead of repeating an extensive list of available literature on the topic, we tried to keep the number of references concise. Hence, we focused on those works that deal with estimation of subsurface properties and that influenced the manuscript.
We added some references (P2 L30ff).

*Page 3, line 7: Replace "may even" with "lead to better convergence and may even"*
*Page 3, line 16 (and many other places. It should be "sensitive to", not "sensitive on".*
*Page 3, line 19: I know that both uses are correct, but I prefer to treat data in plural form: One datum, several data.*
*Page 3, line 12: Add "for porous media" before "the standard".*
*Page 4, line 18: Explain already here that the reason for ignoring the large drainage event is that Richards equation is solved in 1-D.*
*Page 6, line 8: Why only one antenna? Why not model this as a 2-D system?*
**Reply:**
We revised the manuscript accordingly.

*Page 6, line 15: Why conductivity at dc conditions. It should be the conductivity at around 400 MHz, typically 50% or so higher than the DC value.*
**Reply:**
Please refer to the reply to comment number 2 of referee 2.

*Page 6, line 20: Clarify that one normally has no idea about the power of the GPR source, only some basic idea about its shape. This implies that some sort of normalization of observed and simulated traces are needed.*
**Reply:**
We revised the manuscript accordingly (P9 L5f).

*Page 6, equation 7. Here, the dependence of temperature is included, but it is written later that this was not done and it led to errors on the scale of one standard deviation in time. Also, how are boundary conditions in the tank modeled in the EM code?*
**Reply:**
In a field study, the distribution of soil temperature distribution is typically unknown. Hence, we inserted an estimate for the mean soil temperature. However, the error in the signal travel time can be calculated for a given travel path length, e.g., depending on the water content and the error in soil temperature. For the given model, the resulting error in the signal travel time exceeds one standard deviation of the signal travel time already for a deviation of a few Kelvin.
Perfectly matched layers are used as boundary conditions for the electromagnetic model (Sect. 3.1.1).

*Page 6, line 31: Use Archie to explain how big this approximation is. Depends on the differences in porosity of the sands used.*
**Reply:**
Besides the soil porosity, Archie's law (e.g., Friedman, 2005) depends on further parameters that are unknown a priori and may also vary in the subsurface architecture. We merely have estimates from TDR measurements for the electrical conductivity of the bulk. These vary during the experiment approximately between $1 \cdot 10^{-3}$ and $2 \cdot 10^{-2}$ S m$^{-1}$.

*Page 7, line 4: Should be "corresponding" instead of "corresponding".*
*Page 7, lines 13-14: Confusing as treatment to simulated and observed data are mixed. For example, (ii) only important for real data and (iii) only needed for simulated data. Please clarify what is done for (1) simulated data and (2) observed data.*
**Reply:**
We revised the section 2.3 accordingly and added Fig. 3 for clarification.

*Page 7, line 18: Is this correction valid for a dipole radiation pattern. Page 7, line 23: "i" in italics.*
**Reply:**
Yes, this correction is valid for a dipole radiation pattern (see, e.g., Buchner (2012), p. 44ff).
We revised the manuscript accordingly (P8 L3).

*Page 27, line 19: Replace "favourably" with "reasonably well".*
**Reply:**
We revised the section 4 and don't use that sentence anymore.

**References:**

Buchner, J. S., Wollschläger, U., Roth, K. (2012). Inverting surface GPR data using FDTD simulation and automatic detection of reflections to estimate subsurface water content and geometry. Geophysics, 77(4), H45-H55.

Buchner, J. S. (2012). Constructive Inversion of Vadose Zone GPR Observations (Doctoral dissertation), Heidelberg University, `http://archiv.ub.uni-heidelberg.de/volltextserver/14171/4/Buchner_JS_Dissertation_2012.pdf`.

Dagenbach, A., Buchner, J. S., Klenk, P., Roth, K. (2013). Identifying a parameterisation of the soil water retention curve from on-ground GPR measurements. Hydrology and Earth System Sciences, 17(2), 611.

Friedman, S. P. (2005). Soil properties influencing apparent electrical conductivity: a review. Computers and electronics in agriculture, 46(1), 45-70.

Jaumann, S., Roth, K. (2017). Effect of unrepresented model errors on estimated soil hydraulic material properties. Hydrology and Earth System Sciences, 21(9), 4301.

**Reply to referee comment 2**

*The authors deduce subsurface hydraulic properties by an inversion of time-lapse surface GPR measurements during an imbibition experiment of an artificial test site. The coupled inversion process includes a hydraulic simulation by solving the 1D Richards equation and a simulation of radar wave propagation by 2D finite-differences calculation. Water content distribution and electromagnetic soil properties are coupled by a petrophysical relation (CRIM). During the inversion, the misfit between events, i.e. traveltimes and amplitudes of selected reflections, in experimental and synthetic GPR data is minimised. The authors use an inversion scheme that combines several optimisation steps including global and gradient techniques. The approach is first demonstrated for synthetic data and later for experimental data. The result is a 1D subsurface model and for both predefined layers the characteristic hydraulic properties of a Brooks-Corey parameterisation of the water retention function is fitted. The presented work is a relevant contribution towards a non-destructive hydraulic characterisation of the subsurface, which is still an unsolved problem for the unsaturated zone.*

**Reply:** We thank the reviewer for the constructive comments and suggestions. We revised the manuscript accordingly and refer to the revised version in the following.

*However, the manuscript has to be overworked as the whole analysis including GPR data processing and inversion is somehow nebulous and difficult to follow. I would also suggest to shorten the text by writing more tersely, avoiding repetitions and possibly moving some parts into an appendix as e.g. GPR data conversion due to Bleistein, details on event detection/association and inversion. Besides this, some major points have to be clarified: 1. Amplitude handling: The formula used for spherical divergence correction for 3D data (P 7 Eq. 10) seems not correct. Correcting with square root of distance is used for 2D data. Also the dimensions of Eq. 10 do not fit. Various formulations of adequate gain functions for 3D (experimental) data is given in Yilmaz: Seismic Data Analysis (2001), e.g. Eq. (1-8a) $g(t) = \frac{v^2(t)t}{v_0^2 t_0}$. The whole amplitude balancing in the manuscript is not clear to me. The radar traces are normalised several times (P8 L1-2) and normalisation is done relative to the maximal absolute amplitude, which is the first reflection. Is the reflector characteristics constant during the entire experiment? What is the advantage of the complicated amplitude adaption due to Bleistein (1986) compared to a simple correction of 2D circular divergence with the square root of the distance? I would suggest*

*to provide a flow chart of amplitude handling for both experimental (3D) and synthetic (2D) radar data. It seems you use different amplitude handling for event detection and the inversion process?*

**Reply:**

We revised the manuscript, however we decided not to move the GPR evaluation and optimization methods to the appendix since these are actually the essential parts of the manuscript.

Besides an amplitude correction, Bleistein (1986) also provides a correction for the signal frequency. Hence, we use this correction method.

The wave equation is typically transferred to the Helmholtz equation which may be solved with a Green's function approach. To yield the electric field, the resulting Green's function $G$ is convoluted with the function $f$ which is essentially the temporal partial derivative of the source current density ($f \propto \mu \partial_t J$):

$$\widehat{E}(\vec{x}, \omega) = \int \mathrm{d}\vec{x}' \ G(\vec{x}, \vec{x}', \omega) \ f(\vec{x}', \omega). \tag{1}$$

Similar to Bleistein (1986), also other authors, e.g., Miksat et al. (2008), propose Green's functions for a 3D point source and a 3D line source (in $x$-direction) which corresponds to a 2D point source. Theses Green's functions may be transferred into each other in the frequency domain using a correction factor $C_\omega(\omega)$ via

$$\widehat{G}^{\mathrm{3D}}(\vec{x}, \vec{x}', \omega) = \widehat{G}^{\mathrm{2D}}(\vec{x}, \vec{x}', \omega) \cdot C_\omega(\omega). \tag{2}$$

This correction factor is given by $C_\omega(\omega) = \sqrt{\frac{|\omega|}{2\pi\sigma_c}} \exp\left(-\frac{\mathrm{i}\pi}{4}\mathrm{sign}(\omega)\right)$, where $\sigma_c$ denotes the integral of the velocity with respect to the length $s$ of the ray trajectory $\sigma_c = \int c(s)\mathrm{d}s$. Since this correction factor is spatially constant, it may also be used to directly scale the Fourier transform of the electric field:

$$\widehat{E}^{\mathrm{3D}} = \int \mathrm{d}x\mathrm{d}y\mathrm{d}z \ \widehat{G}^{\mathrm{3D}} \ f \tag{3}$$

$$= \int \mathrm{d}x\mathrm{d}y\mathrm{d}z \ \widehat{G}^{\mathrm{2D}} \ C_\omega(\omega) \ f \tag{4}$$

$$= C_\omega(\omega) \int \mathrm{d}x \int \mathrm{d}y\mathrm{d}z \ \widehat{G}^{\mathrm{2D}} \ f \tag{5}$$

$$= C_\omega(\omega) \int \mathrm{d}x \ \widehat{E}^{\mathrm{2D}} \tag{6}$$

$$= C_\omega(\omega) \ C_i \ \widehat{E}^{\mathrm{2D}}. \tag{7}$$

This exploits that (i) the wave propagation is a linear problem in order to separate $C_\omega(\omega)$ and (ii) that the shape of the wave does not change in $x$-direction due to symmetry. Thus, the integration over the $x$-direction leads to the constant $C_i$ (m). This the constant is independent of the frequency and hence does not change the electric field in the inverse Fourier transformation. Therefore, it is possible to directly scale the Fourier transform of the electric field with $C_\omega$ and to use the normalized amplitude of

the electrical field in the space domain. Thus, the value of the constant $C_i$ is irrelevant. By separating the frequency and amplitude correction in the previous version of the manuscript, we set $\sigma_c = 1$ in the frequency correction and applied the correct value in the amplitude correction. We clarified section 2.1 as well as the normalization in the revised version of the manuscript (P9 L4ff, P9 L15f, P10 L4f, P10 L21f).

The first reflection is not always the one with the highest amplitude (see Figs. 13b and 16b – if the event has the maximal amplitude in both the simulation and the measurement, it has no error as both values are equal to 1, thus you can check that the event with maximal absolute amplitude is not always in the same reflection). The characteristics of the first reflector does change over the course of the experiment, due to the hydraulic dynamics (e.g., see marker (3) at Fig. 9).

A flowchart was added in the revised version of the manuscript (Fig. 3). Besides the 2D to 3D conversion and the event selection, simulated and measured data are treated the same.

*2. Neglecting dielectric losses. I'm wondering if at frequencies of about 400 MHz, the impact of free water relaxation can be neglected and whether the imaginary part of permittivity has to be taken into account. When using complex permittivity of water according to Kaatze et al. (1989) and the CRIM formula and a DC conductivity of 0.003 S/m this results in: 3 dB/m (2 dB/m from free water relaxation, 1 dB/m from DC conductivity) for 10vol% water content and 5 dB/m (4 dB/m from free water relaxation and 1 dB/m from DC conductivity) for full saturation (40vol% water content). This means that up to 80% of total loss is caused by polarisation effects of free water. Neglecting these effects results in wrong amplitudes of the simulated data and I'm wondering how they can fit to the field data. At the end of the imbibition experiment (water table at -0.6 m) the amplitudes of the lower reflections (1 m saturated material above, i.e. 2 m two-way travel path) should appear to be approximately 8 dB (2.5 times) higher in the synthetic data than in the field data. I suggest to use the true complex permittivity of water at 400 MHz or, if the FDTD code cannot handle complex property values, an effective HF conductivity including both, DC conduction losses and HF polarisation losses.*

**Reply:**
Using the parameters of Kaatze et. al. (1989) and the measurements of Light et. al. (2005) for the direct current conductivity yields the temperature and frequency dependency of the electrical conductivity of pure water shown in Fig. 1 of this reply. Due to the finite measurement time of the TDR traces, they yield an effective estimate of the electrical conductivity which is larger than the direct current conductivity. Hence, we corrected the notation in the revised version of the manuscript (e.g., P6 L13).

*3. GPR forward calculation: Why is a 2D FDTD code used for a horizontally layered model? A 1D reflectivity method as e.g. used by Bradford et al., (2014) or a 1D FDTD code would be much more efficient. The power of FDTD is certainly that it can be used for complicated 2D/3D subsurface models and thus for inverting 2D/3D data with an according hydraulic simulation. However, in the presented study only 1D data are used and no outlook is given how to adopt the strategy to 2D or 3D problems. From this it is*

[Figure]

Figure 1: The temperature and frequency dependency of pure water using the parameters of Kaatze et. al. (1989) and the measurements of Light et. al. (2005) for the direct current conductivity.

*not clear why the expensive 2D FDTD algorithm is used. The source wavelet of the simulation is different to the wavelet of the experimental data. When dealing with gradient interfaces as the capillary transition zone, the wavelet shape may have a big impact on the maximal amplitude of the reflected signal. Why not using the first reflected signal, which is used for normalisation, as source wavelet in the simulation?*

**Reply:**

In a 1D model, the shape of the wavelet would change, increasing the deviation to the measured 3D antenna signal. Also, using a 1D model would lead to a depth-dependent error in the ray travel path (Fig. 2 of this reply).

We added an outlook on the further application of the proposed algorithm in the revised version of the manuscript (P30 L27ff and P30 L30ff).

Concerning the estimation of the source wavelet, please also note the reply to referee comment 1, point 10.

*Inversion. The complex inversion scheme is a nesting of global and gradient methods. It is somewhat nebulous and it's difficult to get an impression of the quality of original data fit. Why do you use different but relatively narrow boundaries (fit ranges) for the inversion parameters of the two layers? By doing this, the inversion result is biased by a-priori information that is usually not known but the actual aim of the investigation. The inversion should work with the same (broader) fit range for both layers. If not, it cannot be adopted to the field. The fit ranges should be used to provide outer boundaries of the deduced material properties in Fig. 9 and Fig. 14. I'm also missing a figure showing experimental GPR data traces and synthetic traces based on the inversion models to prove that the experimental data are well described. This figure should include the resulting synthetic radar traces of the ten best inversion results to get an idea of the*

[Figure]

Figure 2: (a) The difference in travel path for 1D and 2D, and (b) the error in travelpath

*fluctuations and an idea of the fitting quality.*

**Reply:**

The global-local approach is a common method. It is also typical that larger problems are approached with a preconditioning step.

The deviation in amplitude and travel time as well as the residuals are given in the Figs. 13 and 16.

The fit ranges cover sandy materials. In a field application of the method, the material type of the subsurface can be sampled with a geological drill, e.g., using a "Pürckhauer". The proposed method is intended to be used in combination with published multichannel method Gerhards et. al (2008) and Buchner et. al. (2012) which provide the architecture structure, layer depth, and average water content. These methods have been shown in 2D.

The applied optimization methods use a uniformly distributed prior information. Hence, even if the prior information was included in the cost function, the fit parameter range would not bias the parameter estimation. If the parameter range was too small, the resulting parameters would be close to the boundary. This is not the case in this study. Choosing single traces out of the time-lapse radargram with many traces does not suffice to proof the quality of the fit, because, while the fit might be good for one trace, it could be very bad for other traces. At least in the synthetic study, where there are no additional reflections from the walls and compaction interfaces, the true and the estimated radargrams are very similar. It would be difficult to discern them visually. Thus, we show the evaluated events, their deviations in signal travel time and amplitude as well as the according residuals. This approach allows to pinpoint deviations of the simulation and the measurement very precisely. Hence, showing the 10 best members would require to show 10 plots such as Fig. 13 per study.

*5. In the analysis, the reflection of the compaction layer is excluded. If this interface*

[Figure]

Figure 3: Sketch of the influence of a compaction layer (green) on the water content distribution (blue). Main uncertainties about the shape of the influence are indicated with arrows (1, 2, 3).

*causes a GPR reflection, this must be caused by different water contents on both sides and hence, there must be significant differences in the material hydraulic properties (see P24 L25ff). So why should I ignore an interface that is present in the subsurface and reflects changes in hydraulic properties? Please explain.*

**Reply:**

In order to describe the influence of the compaction interface on the water content distribution quantitatively, at least three uncertainties would have to be estimated (Fig. 3 in this reply): The vertical position of the compaction interface (1), the change of the pore-size distribution at the compaction interface (2) as well as the change of the pore-size distribution with increasing distance from the compaction interface (3).

In vertical soil samples taken at ASSESS, e.g., with a Pürckhauer, we could only visually discern the different sands but no compaction interfaces. Since the quantitative influence of the compaction interfaces on the hydraulic dynamics is unknown a priori, we assume homogeneous material properties in this first step, in particular to investigate the necessity for a detailed quantitative analysis of compaction interfaces. Relevance of this representation error is indicated by the structural residuals after the inversion. The results of this study, i.e. the effect on the estimated parameters and the remaining residuals, suggest that the representation of the compaction interfaces in ASSESS is relevant. Hence, we propose that the effect of all relevant representation errors on the estimated properties should be analyzed in a next step, similar as has been done for TDR data by Jaumann and Roth (2017). This is a significant effort well beyond the scope of this paper.

*6. The title is misleading, I would suggest to delete "...and subsurface architecture. . . " as this would imply at least a 2D subsurface model. The section headings of 2.2 sound unusual to me. From a geophysical perspective the following headings would give a better description: 2.2.1 Water dynamics, 2.2.2 Hydraulic material characterisation 2.2.3*

*Time lapse experiment 2.2.4 GPR investigation and electromagnetic material characterisation.*
**Reply:**
We changed the title to "…and layered architecture…" making it more precise. We clarified the introduction of the representation and thus the titles in the revised version of the manuscript (P4 L19ff).

**Further comments:**

*(P2 L27) References are a bit biased by the own workgroup. E.g., when introducing the FDTD method I would expect the basic work of Yee, Taflove. . . and, e.g., the former ETHZ geophysics group or from the gprMax developers.*
**Reply:**
Instead of repeating an extensive list of available literature on the topic, we tried to keep the number of references concise. Hence, we focused on those works that deal with estimation of subsurface properties and that influenced the manuscript. Still, we agree that classical work on methods should be acknowledged and added the references accordingly (P2 L30ff).

*(P6 L23ff) The CRIM formula uses the square root of permittivities (see your original reference: Birchak et al., 1974). There is no need to first define a general formulation with an exponent $\alpha$, which is not the original CRIM formula, and then fix the exponent $\alpha = 0.5$. Keep it simple and use the square root from the beginning.*
**Reply:**
We revised the manuscript accordingly (P7 L3).

Further comments *(P6 L9) You should describe that you use the static permittivity of water (which is acceptable for 400 MHz, at least for the real part of permittivity).*
**Reply:**
We revised the manuscript accordingly (P6 L16f and P7 L7ff).

*(P7 L14) "…removal of the direct and trailing signal". What is the trailing signal? Is this the interference of ground wave, crosstalk, reflection at the ground surface and the antenna metal shielding? In Fig. 3, a part of this trailing signal is remaining, which is confusing. Why not muting this part?*
**Reply:**
We clarified the paragraph accordingly (P7 L21ff).

*(P7 L15) ". . . we pick the direct signal and subtract it from the radargram" is confusing. Not the signal is subtracted but the travel time.*
**Reply:**
We revised the manuscript accordingly (P7 L24f).

*(P8 L5) "normalized amplitude (original amplitude)". Rephrase, as the amplitude is*

*either normalized or original.*

**Reply:**

We rephrased the section 2.3.2 accordingly.

*(P8 L6) "amplitude is amplified quadratically with travel time" means they are corrected for spherical divergence twice consecutively? Is this just an arbitrary gain function that showed to work well and to correct for spherical and intrinsic attenuation at the specific site? Please explain.*

**Reply:**

This is an arbitrary gain function that showed to work well for the detection of events at lower travel times. This gain function is merely used for the detection, the travel time and amplitude. We clarified the manuscript accordingly (P9 L8ff).

*(P11 L3) Eq. 11: I think the expression has to be divided by M to get the classical $\chi^2$ with $\chi^2 = 1$ if the data are described within the error.*

**Reply:**

Since the number of events (M) changes over the course of the optimization, this would lead to a balancing of the number of associated events and the associated residuals. Hence, the association of two events would only be added, if their residual is smaller than the average residual (what is very unlikely). Thus, the optimization algorithm would tend to decrease the number of associated events in order to decrease the cost.

*(P11 L4) How is the standard deviation of the normalised travel times and amplitudes calculated, i.e. what are the input data?*

**Reply:**

We added this information to the revised manuscript (P17 L6ff).

*(P13 L33) ". . . infinite dipole pointing in x dimension". This should be y dimension (into the plane of projection). Please provide x,y direction in Fig. 6.*

**Reply:**

We clarified the dimensions by adding them to the labels of Fig. 7.

*(P13 L34) a Ricker function is the second derivative of a Gauss-fct, not the first derivative*

**Reply:**

We improved the sentence (P14 L22f).

*(P18 L26) What is the meaning of amplitude information of a single channel? You are using a single channel GPR system and only one antenna, so this expression is confusing.*

**Reply:**

We clarified the paragraph (P23 L1ff).

*(P24 L30) Couldn't the uncertainty of the groundwater table relative to the ground sur-*

*face be overcome by simple levelling the ground surface?*
**Reply:**
In principle, this was possible. However, the bottom and the surface ASSESS site is inclined relative to the groundwater level (approx. 0.1 m over the length of the site (Jaumann and Roth, 2017)). Yet, when applying the method in the field, the uncertainty of the position of the groundwater level is also likely to increase with the distance from the well.

*(P26 L20ff) is a partial repetition of (P22 L5ff (the lower line 5)).*
*(P26 L26) ". . . and that (ii) the direct electric conductivity can be assessed with GPR measurements". I cannot understand the context.*
*(P27 L13ff) This is again a partial repetition of (P22 L5ff) and (P26 L20ff)*
**Reply:**
We revised the section by deleting the short summary (P27, L27ff).

*(P27 L18) Better use "constant offset" (CO) instead of "single-channel" GPR data.*
**Reply:**
In this sentence, we differentiate between single-channel and multi-channel approaches, e.g., used by Buchner et. al. (2012).

*(Fig. 3, caption) Are these synthetic or experimental data?*
**Reply:**
These are simulated data. We clarified the caption of Fig. 4 accordingly.

*Fig. 6: Which E-field component is shown, what is the x and y direction?*
**Reply:**
The x-component of the E-field is shown. We directions are now given in the labels of Fig. 7.

*Fig. 12, bottom: y-axis label: standardized residual: Does 10 mean that the residual is 10 times the STD or should it be 10*
**Reply:**
We clarified the caption of Figs. 13 and 16. It is 10 times the standard deviation, which is for the signal travel time $6 \cdot 10^{-4} \cdot 60$ ns $= 0.36$ ns. Hence, 10 times the standard deviation corresponds to 3.6 ns.

*Fig. 13: I suggest to split the figure into two individual figures as it might be very confusing to mix the 2D radar section with the time lapse data at one location. The label for the groundwater table reflection is "I" in the upper radar section and "2" in the lower time-lapse data. Actually, it's very hard to distinguish the label "I" from the label "1" in the upper radar section. Please use the same and distinct labels for the GWT reflection in all figures.*
**Reply:**
Especially for people that are not used to time-lapse radargrams, having a corresponding common offset radargram of the initial state helps to associate the reflections and to understand their temporal evolution.

We clarified the caption of Fig. 14 and increased the size of the markers. Since the groundwater table is fluctuating, Arabic numbers indicate the water induced reflections at different times. Thus, the labels are used consistently in all the figures.

[revised manuscript text omitted]
. Using the association of the events regularizes the optimization. We apply a global-local optimization approach with preconditioning. For this purpose, we draw parameter sets with the Latin hypercube algorithm . These parameter sets that serve as initial parameters for the preconditioning step. In this step, the simulated annealing algorithm which is sequentially coupled with the and Levenberg–Marquardt algorithm are sequentially coupled allowing only a limited number of iterations working on a subsampled data set. Subsequently, the resulting parameters of the preconditioning step serve as initial parameters for another run of the Levenberg–Marquardt algorithm, now working on the full data set. We show that this procedure allows to accurately estimate the subsurface architecture and the associated effective hydraulic material properties for synthetic and measurement data.

**2 Methods**

**2.1 ASSESS**

The ASSESS test site measurement data for this work are acquired at an approximately $2\,\text{m} \times 20\,\text{m} \times 4\,\text{m}$ large the test site (ASSESS) which is located near Heidelberg, Germany, and consists of three different kinds of sand (materials A, B, and C). Its effective provides an effectively 2D subsurface architecture is visualized in subsurface architecture consisting of three kinds of sands (Fig. 1. The approximately $2\,\text{m} \times 20\,\text{m} \times 4\,\text{m}$ large site is equipped with a well to monitor and manipulate the groundwater table, a weatherstation, a tensiometer (UMS T4-191), as well as 32 soil temperature and TDR sensors. A geotextile separates the sand from an approximately ). Below the sands, a $0.1\,\text{m}$ thick gravel layer below, which ensures a rapid water pressure distribution ensures rapid distribution of the water pressure at the lower boundary. This gravel layer is separated from the sand via a geotextile and is the only connection of the groundwater well to the rest of the test site. Below this gravel layer, site to a groundwater well. The groundwater well is in particular used to manipulate the groundwater table by pumping water in and out of the well. The groundwater table is measured automatically with a tensiometer (UMS T4-191)

[Figure]

**Figure 1.** ASSESS  provides an  effectively 2D geometry with subsurface architecture consisting of 
[revised manuscript text omitted]

**3 Application**

In this section, we apply the presented methods to the acquired are applied to GPR data. Therefore, we first explain To this end, the setup of the case studyand the , its implementation and the detailed setup of the parameter estimation procedure

are explained first (Sect. 3.1.1). Then, we test the method with synthetic data to understand the phenomenology of the data and capabilities of the method (Sect. 3.1). Subsequently, the suitability of the presented methods to estimate the subsurface architecture and the corresponding soil hydraulic material properties is first tested with synthetic data 3.2). Finally, we apply the method to the measured the methods are applied to measurement data (Sect. 3.3)and analyze the accuracy of the resulting parameters.

**3.1 Setup of the case study**

**3.1.1 Implementation**

For the simulation of the GPR signal of antenna 3, we assume a layered subsurface architecture (Fig. 1). The transmitter of the antenna is represented with an infinitesimal dipole (t) and the electric field is read at the position of the receiver antenna (r). An absorbing layer is used as boundary condition.

The numerical solution of the

The Richards equation (Eq. 1) is based on solved numerically with $\mu\varphi$ (muPhi, Ippisch et al., 2006) which uses a cell centered finite volume scheme with full upwinding in space and an implicit Euler scheme in time. The nonlinear equations are linearized by an inexact Newton method with line search and the linear equations are solved with an algebraic multigrid solver. We solve the Richards equation in 1D ($z$dimension-dimension) on a structured grid with a resolution of $\approx 0.005$ m.

Generally, the boundary condition is implemented with a Neumann no-flow condition. However, during the forcing phases, we prescribe the measured groundwater table as a Dirichlet boundary condition at the position of the groundwater well. We initialize the simulation with hydraulic equilibrium based on the measured groundwater table position.

The simulated water content is converted to relative permittivity via the CRIM using the mean soil temperature $T_\mathrm{s} = 8.5\ {}^\circ\mathrm{C}$ (Sect. 2.2.4).

To simulate the temporal propagation of the electromagnetic signal, we solve Maxwell's equations (Sect. 2.2.4) in 2D with the MIT electromagnetic equation propagation software (MEEP, Oskooi et al., 2010). The transmitter antenna is represented with an infinite dipole pointing in $x$dimension-dimension. Thus, we neglect any effects from the real antenna geometry (bow tie), cross coupling or antenna shielding. The antenna source current density $\boldsymbol{J}$ is given by a Ricker an excitation function (first derivative of a Gaussian–shaped function) that leads to a Ricker wavelet with a center frequency of $400$ MHz. The receiver antenna is not represented explicitly. Instead, $E_x$ is read directly at the position of the receiver antenna. We use the antenna separation of the real GPR system ($0.14$ m) in the simulation. Perfectly matched layers (PML) of $0.15$ m thickness serve as boundary condition. The initial electromagnetic field in the domain is zero.

We use one tenth of the minimal wavelength $\lambda_\mathrm{w,min}$ as upper limit for the spatial resolution $\Delta z$:

$$\Delta z \leq \frac{\lambda_\mathrm{w,min}}{10} = \frac{\frac{c_0}{\sqrt{\varepsilon_\mathrm{r,max}}}}{10 f_\mathrm{max}} \approx 0.007\ \mathrm{m}, \tag{15}$$

[Figure]

**Figure 7.** For the simulation of the GPR signal, we assume a layered subsurface architecture (Fig. 1). The transmitter of the antenna is represented with an infinitesimal dipole (T) and the electric field is read at the position of the receiver antenna (R). An absorbing layer is used as boundary condition. The $x$-component resulting electric field after $5$ ns is shown. The markers for the material interfaces are used consistently in this work. This figure only shows the electromagnetic simulation. The hydraulic simulation is not shown here, such that the subsurface architecture can be defined clearly. 
[revised manuscript text omitted]
_0$ $(\mathrm{m\,s^{-1}})$ $K_s$ $(\mathrm{m\,s^{-1}})$ | $10^{-3.5}$ | $10^{-3.4 \pm 0.2}$ |
| | $\tau$ $(-)$ | $0.5$ | $0.6 \pm 0.2$ |
| | $\theta_\mathrm{s}$ $(-)$ | $0.38$ | $0.38 \pm 0.01$ |
| | $\theta_\mathrm{r}$ $(-)$ | $0.03$ | $0.027 \pm 0.006$ |
| A | $h_0$ (m) | $-0.20$ | $-0.199 \pm 0.008$ |
| | $\lambda$ $(-)$ | $2.5$ | $2.8 \pm 0.7$ |
| | $K_0$ $(\mathrm{m\,s^{-1}})$ $K_s$ $(\mathrm{m\,s^{-1}})$ | $10^{-4.5}$ | $10^{-4.47 \pm 0.05}$ |
| | $\tau$ $(-)$ | $0.5$ | $0.4 \pm 0.5$ |
| | $\theta_\mathrm{s}$ $(-)$ | $0.41$ | $0.41 \pm 0.02$ |
| | $\theta_\mathrm{r}$ $(-)$ | $0.05$ | $0.06 \pm 0.02$ |
| Gravel | $\theta_\mathrm{s}$ $(-)$ | $0.40$ | $0.40 \pm 0.03$ |
| Architecture | $h_1$ (m) $d^\mathrm{V}$ (m) | $1.00$ | $0.99 \pm 0.02$ |
| | $h_2$ (m) $d^\mathrm{VI}$ (m) | $0.20$ | $0.20 \pm 0.01$ |

**Table 3.** The mean and the standard deviation are calculated using the resulting parameters from the ten best ensemble members (Sect. 3.1.2) estimated from measured data. The corresponding material functions are  shown in Fig. 15. The reference parameters for the materials A and C are determined from TDR data acquired during the same experiment  (Jaumann and Roth, 2017). Note that the according standard deviations  are determined  from a single Levenberg–Marquardt run and  thus are only representative for one local minimum. Also, these standard deviations are given with the understanding that they are specific to the applied algorithm and will change for different algorithm parameters. Hence, these standard deviations are in particular not suitable to compare the precision of the TDR and GPR evaluation. Notice that for the TDR evaluation the porosity of the materials is assumed to be known from core samples. The reference parameters for the subsurface architecture are calculated from ground truth measurements acquired during the construction of ASSESS. The corresponding standard deviations are given according to Buchner et al. (2012).

| Material | Parameter | Reference | Mean results |
|---|---|---|---|
| C | $h_0$ (m) | $-0.159 \pm 0.004$ | $-0.13 \pm 0.03$ |
| | $\lambda$ $(-)$ | $3.28 \pm 0.02$ | $3.3 \pm 0.7$ |
| | $K_0$ $(\mathrm{m\,s^{-1}})$ $K_s$ $(\mathrm{m\,s^{-1}})$ | $10^{-3.70 \pm 0.02}$ | $10^{-3.6 \pm 0.3}$ |
| | $\tau$ $(-)$ | $0.74 \pm 0.06$ | $1.4 \pm 0.4$ |
| | $\theta_s$ $(-)$ | $0.38$ | $0.38 \pm 0.01$ |
| | $\theta_r$ $(-)$ | $0.026 \pm 0.002$ | $0.071 \pm 0.005$ |
| A | $h_0$ (m) | $-0.184 \pm 0.005$ | $-0.20 \pm 0.03$ |
| | $\lambda$ $(-)$ | $1.94 \pm 0.07$ | $2.1 \pm 0.7$ |
| | $K_0$ $(\mathrm{m\,s^{-1}})$ $K_s$ $(\mathrm{m\,s^{-1}})$ | $10^{-4.212 \pm 0.004}$ | $10^{-4.5 \pm 0.1}$ |
| | $\tau$ $(-)$ | $0.33 \pm 0.07$ | $0.4 \pm 1.0$ |
| | $\theta_s$ $(-)$ | $0.41$ | $0.40 \pm 0.01$ |
| | $\theta_r$ $(-)$ | $0.025 \pm 0.004$ | $0.07 \pm 0.03$ |
| Gravel | $\theta_s$ $(-)$ | | $0.41 \pm 0.02$ |
| Architecture | $h_1$ (m) $d^V$ (m) | $0.99 \pm 0.05$ | $0.97 \pm 0.02$ |
| | $h_2$ (m) $d^{VI}$ (m) | $0.13 \pm 0.05$ | $0.17 \pm 0.02$ |